# 3D PRE-TRAINING IMPROVES GNNS
# FOR MOLECULAR PROPERTY PREDICTION

## ABSTRACT

Molecular property prediction is one of the fastest-growing applications of deep learning with critical real-world impacts. Including 3D molecular structure as input to learned models improves their performance for many molecular tasks. However, this information is infeasible to compute at the scale required by several real-world applications. We propose pre-training a model to reason about the geometry of molecules given only their 2D molecular graphs. Using methods from self-supervised learning, we maximize the mutual information between 3D summary vectors and the representations of a Graph Neural Network (GNN) such that they contain latent 3D information. During fine-tuning on molecules with unknown geometry, the GNN still produces implicit 3D information and can use it to improve downstream tasks. We show that 3D pre-training provides significant improvements for a wide range of properties, such as a 22% average MAE reduction on eight quantum mechanical properties. Moreover, the learned representations can be effectively transferred between datasets in different molecular spaces.

## 1 INTRODUCTION

The understanding of molecular and quantum chemistry is a rapidly growing area for deep learning with models having direct real-world impacts in quantum chemistry (Dral, 2020), protein structure prediction (Jumper et al., 2021), materials science (Schmidt et al., 2019), and drug discovery (Stokes et al., 2020). In particular, for the task of molecular property prediction, GNNs have had great success (Yang et al., 2019).

GNNs operate on the molecular graph by updating each atom's representation based on the atoms connected to it via covalent bonds. However, these models reason poorly about other important interatomic forces that depend on the atoms' relative positions in space. Previous works showed that using the atoms' 3D coordinates in space improves the accuracy of molecular property prediction (Schütt et al., 2017; Klicpera et al., 2020b; Liu et al., 2021; Klicpera et al., 2021).

However, using classical molecular dynamics simulations to explicitly compute a molecule's geometry before predicting its properties is computationally intractable for many real-world applications. Even recent Machine Learning (ML) methods for conformation generation (Xu et al., 2021b; Shi et al., 2021; Ganea et al., 2021) are still too slow for large-scale applications.

**Our Solution: 3D Infomax** We pre-train a GNN to encode implicit 3D information in its latent vectors using publicly available molecular structures. A GNN is pre-trained by maximizing the mutual information (MI) between its embedding of a 2D molecular graph and a representation capturing the 3D information that is produced by a separate network. This way, the GNN learns to produce latent 3D information using only the information given by the 2D molecular graphs. After pre-training, the weights can be transferred and fine-tuned on molecular datasets where no 3D information is available. For those molecules, the GNN is still able to produce implicit 3D information that can be used to inform property predictions.

Several other self-supervised learning (SSL) methods that do not use 3D information have been proposed and evaluated to pre-train GNNs and obtain better property predictions after fine-tuning (Hu et al., 2020b; You et al., 2020; Xu et al., 2021a). These often rely on augmentations (such as removing atoms) that significantly alter the molecules while assuming that their properties do not

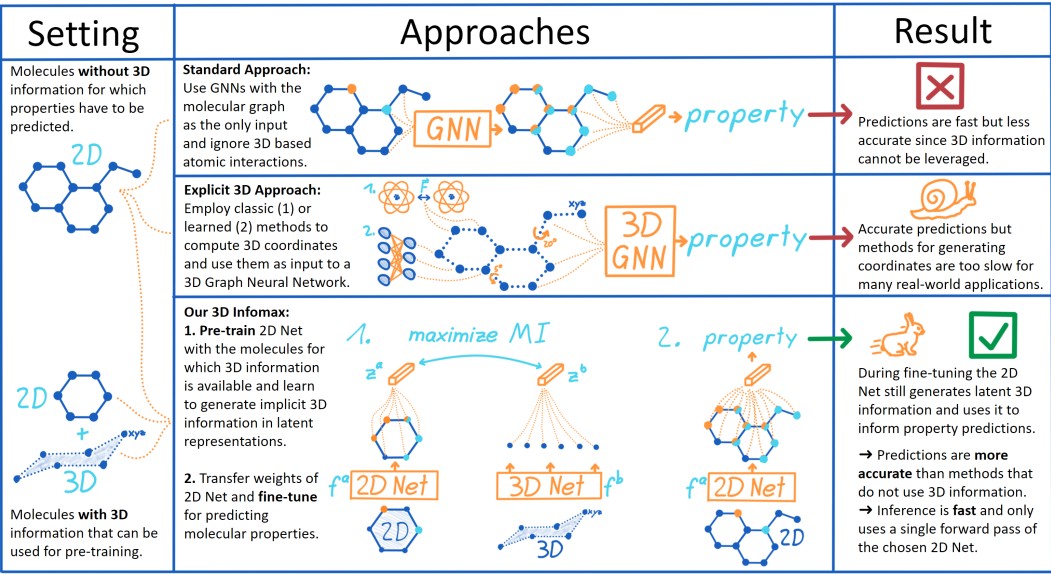

Figure 1: The considered problem setting and the motivation for our 3D Infomax pre-training.

change. Meanwhile, 3D Infomax pre-training teaches the model to reason about how atoms interact in space, which is a principled and generalizable form of information.

We analyze our method's performance by pre-training with multiple 3D datasets before evaluating on ten quantum mechanical properties and ten datasets with biological, pharmacological, or chemical properties. 3D Infomax improves property predictions by large margins and the learned representations are highly generalizable: significant improvements are obtained even when the molecular space of the pre-training dataset is vastly different (e.g., in size) from the kinds of molecules in the downstream tasks. While conventional pre-training methods sometimes suffer from negative transfer (Pan & Yang, 2010), i.e., a decrease in performance, this is not observed for 3D Infomax.

**Our main contributions are:**

- A 3D pre-training method that enables GNNs to reason about the geometry of molecules given only their 2D molecular graphs, which improves property predictions.
- Experiments showing that our learned representations are meaningful for various quantum mechanical, chemical, biological, or pharmacological tasks, without negative transfer.
- Empirical evidence that the embeddings generalize across different molecular spaces.
- An approach to leverage information from multiple molecular conformers that further improves downstream property predictions and an evaluation to what extent this is possible.

## 2 BACKGROUND

**Molecular Conformers** We pre-train models to learn 3D information and transfer it to downstream molecular property prediction tasks. This 3D information is not given by a single set of coordinates. For a given molecular graph there are multiple conformers, i.e., probable arrangements of the atoms, which can lead to different chemical properties. It is usual to consider only the conformers of lower energy since they have a higher probability of naturally occurring.

Several tools exist to compute conformers ranging from methods based on classical force fields to slower but more accurate molecular dynamics simulations. Methods such as RDKit's ETKDG algorithm (Landrum, 2016) are fast but in our experiments we find that their less accurate 3D information does not necessarily improve predictions. The popular metadynamics method CREST (Grimme, 2019) offers a good tradeoff between speed and accuracy but still requires about 6 hours per drug-like molecule per CPU-core (Axelrod & Gomez-Bombarelli, 2020). This highlights the need to capture 3D information without explicitly computing structures, especially for drug-discovery screening datasets comprising of millions or billions of molecules (Gorgulla et al., 2020).

**Symmetries of Molecules** A molecule's conformation does not change if all the atom coordinates are jointly translated or rotated around a point, i.e., molecules are symmetric with respect to these two types of transformations which is also called SE(3) symmetry. Note that some molecules (called chiral) are not invariant to reflections: their properties depend on their chirality. Deep learning architectures that capture these symmetries are usually more sample efficient, and they generalize to all symmetric inputs the architecture has been designed for (Bronstein et al., 2021). In our method, the produced representations of the 3D structure respect these symmetries of molecules.

**Scaffold Split** The preferred way to evaluate molecular models is to use a scaffold split when generating the train-test sets, such that the molecules from the test set do not share a scaffold with those of the training set. This helps to avoid overestimating the generalization power (Yang et al., 2019) since ML models tend to memorize and overfit these structures. We use the common Bemis-Murcko scaffold (Bemis & Murcko, 1996) (see Figure 5 in Appendix A). Molecules that share identical scaffolds are put into the same set, i.e., each scaffold goes into either the train, validation or test set.

**Graph Neural Networks** We make use of GNNs to predict molecular properties given a molecular graph. Many GNNs can be described in the framework of Message Passing Neural Networks (MPNNs) (Gilmer et al., 2017), such as the PNA model (Corso et al., 2020) which we employ.

The aim of MPNNs is to learn a representation of a graph $\mathcal{G} = (\mathcal{V}, \mathcal{E})$ with vertices $\mathcal{V}$ connected by edges $\mathcal{E}$. They do so by iteratively applying message-passing layers and then combining all vertex representations in a readout function. A message-passing layer first creates messages for each edge based on the vertices it connects, then each vertex representation is updated by aggregating the messages of all connected edges and combining them with the previous layer's representation. The messages are usually created by multi-layer perceptrons (MLPs) and are aggregated via permutation invariant functions such as taking their mean, max, or sum. After the message-passing layers, another permutation invariant function is used as *readout* to obtain a final graph level representation.

## 3  RELATED WORK

**Molecular property predictions** While ours is the first work on pre-training GNNs for molecular property prediction using 3D information, it heavily draws from the fields of SSL and ML for molecules. An important milestone for the latter was Gilmer et al. (2017) introducing MPNNs after which GNNs became popular for quantum chemistry (Brockschmidt, 2020; Tang et al., 2020; Withnall et al., 2020), drug discovery (Li et al., 2017; Stokes et al., 2020; Torng & Altman, 2019), and molecular property prediction in general (Coley et al., 2019; Hy et al., 2018; Unke & Meuwly, 2019). The field is well established with easily accessible molecular datasets driving progress (Wu et al., 2017; Hu et al., 2020a) and rigorous evaluations of MPNNs for property prediction (Yang et al., 2019) showing the effectiveness of the approach.

While these GNNs have had great successes by operating on the 2D graph, many tasks on molecules can be improved by additionally using 3D information. A simple approach is to use bond lengths as edge features (Chen et al., 2020a), but methods that capture more molecular geometry improve on this such as SchNet (Schütt et al., 2017). Similarly, DimeNet (Klicpera et al., 2020b;a) proposed extracting more 3D information via bond angles, which further improved quantum mechanical property prediction. Spherical Message Passing (SMP) (Liu et al., 2021) included another angular quantity, and GemNet (Klicpera et al., 2021) developed an approach to also capture torsion angles, such that all relative atom positions are uniquely defined. Equivariant Graph Neural Networks (EGNN) (Satorras et al., 2021) achieved the same by operating on all pairwise atom distances.

**Self-Supervised Learning** attempts to find supervision signals in unlabelled data to learn meaningful representations. In particular, contrastive learning (van den Oord et al., 2018; Gutmann & Hyvärinen, 2010; Belghazi et al., 2018; Hjelm et al., 2019) is a popular class of methods. These learn representations by comparing the embeddings of similar and dissimilar inputs and have achieved impressive results in computer vision (Chen et al., 2020b; Caron et al., 2020).

Learning from unlabeled data also is a critical challenge in molecular chemistry since datasets are relatively small due to experimental costs. Several works have explored contrastive learning variants in the context of molecular graphs for non-quantum molecular properties (Hu et al., 2020b; Wang et al., 2021; You et al., 2020; 2021; Xu et al., 2021a). The improvements these methods provided in molecular property prediction are still limited and often fail to generalize.

# 4 METHOD

Our goal is a method that produces latent 3D information from a 2D molecular graph and use it for more accurate molecular property predictions. For this purpose, we propose to **pre-train** a GNN to produce latent representations of a molecule's 3D geometry. Through this pre-training, the network learns the connection between the 2D information of molecular graphs and their 3D conformers.

After pre-training, we transfer the weights and **fine-tune** them on property prediction tasks as visualized in Figure 2. The GNN's produced 3D information can be used to improve predictions. The following presents our 3D Infomax pre-training before explaining the baselines we compare with.

## 4.1 3D INFOMAX PRE-TRAINING

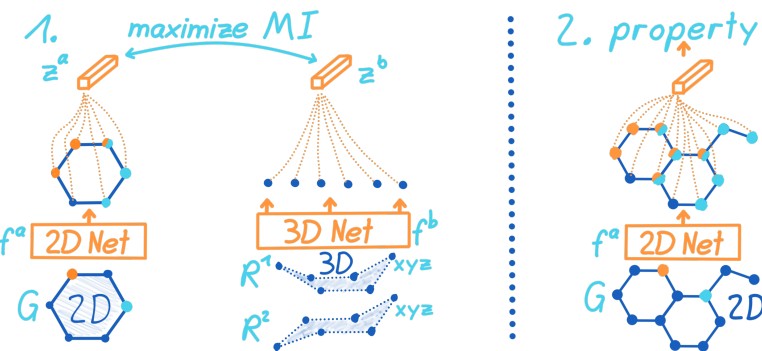

Figure 2: We first pre-train a 2D network $f^a$ by maximizing the mutual information (MI) between its representation $z^a$ of a molecular graph $\mathcal{G}$ and a 3D representation $z^b$ produced from the molecules' conformers $R^j$. In step 2, the weights of $f^a$ are transferred and fine-tuned to predict properties.

3D Infomax uses two different models, as visualized in Figure 2. Firstly, the model that should be pre-trained which we call *2D network $f^a$* since its inputs are 2D molecular graphs $\mathcal{G} = (\mathcal{V}, \mathcal{E})$ with atoms $\mathcal{V}$ and bonds $\mathcal{E}$ from which it produces a representation $f^a(\mathcal{G}) = z^a \in \mathbb{R}^{d_z}$. Secondly, the *3D network $f^b$* which encodes the atoms' 3D coordinates $R = \{r_v\}_{v \in \mathcal{V}}$ in a 3D representation $f^b(R) = z^b \in \mathbb{R}^{d_z}$. Our pre-training can also be understood from a knowledge distillation perspective where the student 2D network learns from the teacher 3D network to produce 3D information.

**Contrastive Framework** To teach the 2D network $f^a$ to produce 3D information from the 2D graph inputs, we maximize the mutual information between the latent 2D representations $z^a$ and 3D representations $z^b$. Intuitively, we wish to maximize the agreement between $z^a$ and $z^b$ if they are derived from the same molecule. For this purpose, we use contrastive learning (visualized in Figure 3). We consider a batch of $N$ molecular graphs $\{\mathcal{G}_i\}_{i \in \{1...N\}}$ with their atom coordinates $\{R_i\}_{i \in \{1...N\}}$ from which the networks produce multiple representations $z_i^a$ and $z_i^b$.

The first objective of contrastive learning is to maximize the representations' similarity if they are a positive pair, meaning that they come from the same molecule (same index $i$). The second objective is to enforce dissimilarity between negative pairs $z_i^a$ and $z_k^b$ where $i \neq k$, i.e., the 2D and 3D representations in the batch should be dissimilar if they come from different molecules. These objectives are captured in the popular NTXent loss (Chen et al., 2020b) and we use a similar loss to jointly optimize our models:

$$\mathcal{L} = -\frac{1}{N} \sum_{i=1}^{N} \left[ log \frac{e^{sim(z_i^a, z_i^b)/\tau}}{\sum_{\substack{k=1 \\ k \neq i}}^{N} e^{sim(z_i^a, z_k^b)/\tau}} \right] \tag{1}$$

where $sim(z^a, z^b) = z^a \cdot z^b / (\|z^a\| \|z^b\|)$ is the cosine similarity and $\tau$ is a temperature parameter which can be seen as weight for the most similar negative pair. While different combinations of contrastive losses and SSL are possible to learn a joint embedding space between 2D and 3D representations, we found the above loss to perform best. Other methods (Barlow Twins (Zbontar et al., 2021), BYOL (Grill et al., 2020), VICReg (Bardes et al., 2021)) are explored in Appendix C.4.

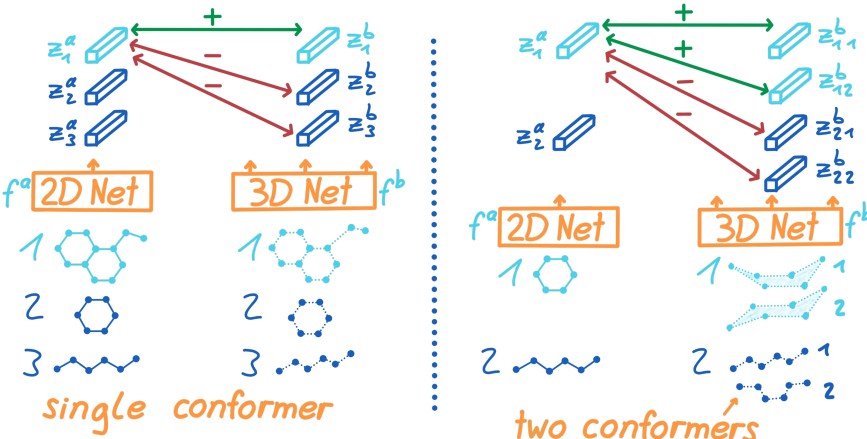

Figure 3: The **single conformer** example shows a batch of three molecular graphs as input to the 2D network with the corresponding three conformers as input to the 3D model. During 3D pre-training, the contrastive loss $\mathcal{L}$ enforces high similarity between latent representations that come from the same molecule (green arrows) while encouraging dissimilarity otherwise (red arrows). This is depicted for the first molecule, but the same is calculated for the second and third. The final loss is the average. The **multiple conformer** example on the right shows two conformers per molecule $c = 2$, and the loss is adjusted to treat all of them as positive pairs if they come from the same molecule and as negative pairs otherwise. Our loss $\mathcal{L}^{multi3D}$ achieves this.

**Multiple Conformers** For most molecules, there are multiple low-energy stable conformers. Instead of only using the most probable conformer (with the lowest energy), we found that leveraging structural information from multiple conformers provides significant benefits. To achieve this, we now consider the $c$ highest probability conformers $\{R_i^j\}_{j \in \{1...c\}}$ of the i-th molecule. If there are fewer than $c$ conformers for a molecule, the lowest energy conformer is repeated. Our choice for the following approach is justified by its good trade-off between simplicity and performance in the comparisons with other possible methods in Appendix C.2.

For every molecule the 3D network now takes all conformers as input and produces their latent 3D representations $\{z_{i,j}^b\}_{j \in \{1...c\}}$. The objective is to maximize the similarity between $z_i^a$ and all conformer representations $z_{i,j}^b$ that stem from the same molecule (see Figure 3). As such, we modify our loss to sum over the similarities of all conformers to obtain the final loss:

$$\mathcal{L}^{multi3D} = -\frac{1}{N} \sum_{i=1}^{N} \left[ log \frac{\sum_{j=1}^{c} e^{sim(z_i^a, z_{i,j}^b)/\tau}}{\sum_{\substack{k=1 \\ k \neq i}}^{N} \sum_{j=1}^{c} e^{sim(z_i^a, z_{k,j}^b)/\tau}} \right]. \tag{2}$$

**3D Network** As the 2D network is an arbitrary GNN that should be pre-trained for which we choose Principal Neighborhood Aggregation (PNA) (Corso et al., 2020), the last missing part of our method is the 3D network. Its purpose is to encode as much 3D information as possible into the vector $z^b$ and it does not have access to the 2D information used by the 2D network such as atom or bond features. Otherwise, the mutual information could be increased by both networks encoding this information instead of the desired 3D information.. Our concrete architecture encodes the 3D information given by the pairwise Euclidean distances of all atoms. This representation uniquely defines all relative atom positions and is invariant to translation and rotation, as desired, but also to reflection, which is a disadvantage since molecular properties can change under reflections (chiral molecules). Using all pairwise distances also means that the method's complexity is quadratic in the number of atoms, but this is feasible for drug-like molecules.

The pairwise distances $d_{uv}$ between atoms $u$ and $v$ are first mapped to a higher dimensional space using high-frequency functions. This motivation for this is to enable deep networks to better fit data with high-frequency variation (Rahaman et al., 2019; Tancik et al., 2020). This scenario is present in our case with small differences in bond lengths. These bond distances and their small variations

might be the most important ones assuming that close-by atoms have the most relevant interactions. As such, we use the following mapping $\gamma : \mathbb{R} \mapsto \mathbb{R}^{2F+1}$ with the number of frequencies $F$ set to 4:

$$\gamma(d_{uv}) = (d_{uv}, sin(d_{uv}/2^0), cos(d_{uv}/2^0), \ldots, sin(d_{uv}/2^{F-1}), cos(d_{uv}/2^{F-1})). \quad (3)$$

The further components can be seen as an MPNN (Gilmer et al., 2017) operating on the fully connected graph of a molecule with the encoded distances as edge features and a constant learned vector as node features. We use these initial node representations instead of atom features (as in the 2D network) such that the mutual information cannot be increased by both networks only encoding the atom feature information. Instead, 3D information has to be captured to solve the objective. The message passing layers iteratively encode the 3D information into the node features, which are pooled to produce the 3D representation $z^b$. The differences to standard MPNNs are detailed in Appendix A. Instead of the presented architecture, a 3D GNN such as SMP (Liu et al., 2021) operating on learned node embeddings could also be used. We justify choosing our architecture with experiments in Appendix C.1 that also show the $\gamma$ mapping's effectiveness by ablating it.

### 4.2 PRE-TRAINING BASELINES

**Distance Predictor** A simpler method to 3D pre-train a GNN instead of 3D Infomax is by directly predicting all atom distances. To predict the distance between node $v$ and $u$, we concatenate their representations $\boldsymbol{h}_u, \boldsymbol{h}_v \in \mathbb{R}^{d_h}$ that were produced by the GNN and feed them to an MLP that produces a single scalar $U : \mathbb{R}^{2d_h} \mapsto \mathbb{R}$. The distance prediction $dist_{uv}$ is then given by

$$dist_{uv} = \text{softplus}(U(\boldsymbol{h}_v \parallel \boldsymbol{h}_u) + U(\boldsymbol{h}_u \parallel \boldsymbol{h}_v)) \quad (4)$$

where $\parallel$ denotes concatenation and $\text{softplus}(x) = log(1 + e^x)$. The node representations are concatenated in both orders and fed to the MLP to ensure that the function is symmetric. The final loss to pre-train $f$ is the mean squared error between the predicted and true distances.

**Conformer Generation** As a second 3D pre-training alternative to 3D Infomax, we pre-train a GNN by generating molecular conformers via the state-of-the-art (SOTA) method GeoMol (Ganea et al., 2021). To predict correct 3D conformers, the GNN has to encode 3D information in its hidden layers which can potentially be transferred and used to inform downstream property predictions.

**GraphCL** We compare against the conventional augmentation-based pre-training method Graph Contrastive Learning (GraphCL) (You et al., 2020) with the settings of JOAO (You et al., 2021) since it outperformed other SSL approaches for multiple molecular tasks. It uses a common self-supervised objective in which the model has to learn to produce representations that are invariant to augmentations. We use randomly dropping nodes with a ratio of 0.2 on both branches of the SSL setup since JOAO found this combination of augmentations to work particularly well for molecules.

## 5 EXPERIMENTS

**Data and Setup** For pre-training, we use three datasets of molecules with 3D conformer information: **QM9** (Ramakrishnan et al., 2014) which contains 134k small molecules (18 atoms on average) with a single conformer, **GEOM-Drugs** (Axelrod & Gomez-Bombarelli, 2020) with 304k molecules and **QMugs** (Isert et al., 2021) with 665k. GEOM-Drugs and QMug, both consist of larger drug-like molecules (44.4 and 30.6 atoms on average) with multiple conformers. For fine-tuning we consider quantum properties on the one side and biological, chemical, and pharmacological properties on the other side. We predict ten quantum properties of QM9 and GEOM-Drugs (a disjoint half of the datasets if the other half was used for pre-training), for which we employ a random split. For non-quantum properties, we use ten Open Graph Benchmark (OGB) (Hu et al., 2020a) datasets with their standard scaffold splits. We use OGB's atom and bond featurization for all datasets. Details for all used data are in Appendix B.1.

We choose PNA (Corso et al., 2020) as the GNN to pre-train due to its simplicity and SOTA performance for molecular tasks. The reported confidence intervals are one standard deviation calculated from six random weight initializations, unless stated otherwise. All baselines we compare with use the same GNN as our 3D Infomax method and all experimental settings are detailed in Appendix B. Code to 3D pre-train a GNN or to reproduce results is available at `https://anonymous.4open.science/r/3141`.

## 5.1 QUANTUM MECHANICAL PROPERTIES

We use 3D Infomax to pre-train three different instances of PNA (1) on 50k molecules from QM9 using a single conformer, (2) on 140k of GEOM-Drugs with 5 conformers and, (3) on 620k of QMugs using 3 conformers. For comparison, we use two different pre-training methods. These are GraphCL (You et al., 2020) as described in Section 4.2 and pre-training by predicting the Gibbs free enery of GEOM-Drugs' pre-training subset (labeled *PropPred*). All pre-training methods use a batch size of 500.

After pre-training, the models are fine-tuned on 50k molecules from QM9 (in Table 1) or 140k from GEOM-Drugs (in Table 2) that have no overlap with the molecules from the pre-training data. On the same molecules, we also train PNA with random weight initialization (labeled *Rand Init*) to compare how much the downstream performance is improved by the different pre-training methods. Furthermore, we train and test the 3D GNN SMP (Liu et al., 2021) on the same molecules with 3D coordinates generated by RDKit's ETKDG algorithm, (Landrum, 2016) which can be done in a fast manner (labeled *RDKit SMP*). Using conformers generated by the SOTA learned method GeoMol (Ganea et al., 2021) always performed worse (Appendix C.6). Lastly, we evaluate SMP using the accurate ground truth 3D conformers of QM9 which were computed with time-consuming simulations that would be infeasible for many real-world applications. These structures are not available to the other methods.

Table 1: Mean Absolute Error (MAE) for QM9's properties. **3D Infomax** is tested with three different pre-training datasets and **GraphCL** uses a two times larger subset of GEOM-Drugs. **True 3D SMP** is a 3D GNN using ground truth 3D coordinates (hidden from other methods). Details on confidence intervals are in Appendix B. Colors indicate improvement (lower MAE) or worse performance compared to the randomly initialized (**Rand Init**) model.

| Target | Rand Init | Pre-training baselines | | Our 3D Infomax | | | RDKit SMP | True 3D SMP |
| | | GraphCL | PropPred | QM9 | Drugs | QMugs | | |
|---|---|---|---|---|---|---|---|---|
| $\mu$ | $0.4133_{\pm 0.003}$ | 0.3937 | 0.3975 | **0.3507** | **0.3512** | 0.3668 | 0.4344 | 0.0726 |
| $\alpha$ | $0.3972_{\pm 0.014}$ | 0.3295 | 0.3732 | 0.3268 | 0.2959 | **0.2807** | 0.3020 | 0.1542 |
| homo | $82.10_{\pm 0.33}$ | 79.57 | 93.11 | **68.96** | 70.78 | 70.77 | 82.51 | 56.19 |
| lumo | $85.72_{\pm 1.62}$ | 80.81 | 99.84 | **69.51** | 71.38 | 78.10 | 80.36 | 43.58 |
| gap | $123.08_{\pm 3.98}$ | 120.08 | 131.99 | **101.71** | 102.59 | 103.85 | 114.24 | 85.10 |
| r2 | $22.14_{\pm 0.21}$ | 21.84 | 29.21 | **17.39** | 18.96 | 18.00 | 22.63 | 1.51 |
| ZPVE | $15.08_{\pm 2.83}$ | 12.39 | 11.17 | 7.966 | 9.677 | 12.06 | **5.18** | 2.69 |
| $c_v$ | $0.1670_{\pm 0.004}$ | 0.1422 | 0.1795 | 0.1306 | 0.1409 | **0.1208** | 0.1419 | 0.0498 |

Table 1 shows that 3D Infomax pre-training leads to large improvements over the randomly initialized baseline and over GraphCL with all three pre-training datasets. After 3D pre-training on one half of QM9, the average decrease in MAE is 22%. Comparing 3D Infomax on GEOM-Drugs with GraphCL shows that even though the latter is pre-trained on two times as many molecules from the same dataset, 3D pre-training is always better by a large margin.

Pre-training with the disjoint half of QM9 performs best since it shares the molecular space of the test set. Nevertheless, the learned representations also generalize well: pre-training on GEOM-Drugs and QMugs leads to improvements of 19% and 18% respectively, even though QM9 contains much smaller molecules with on average 18 atoms compared to the 44.4 atoms for the drug-like molecules of GEOM-Drugs.

Table 2: The MAE for GEOM-Drugs' properties. **3D Infomax** compared with **GraphCL** and no pre-training.

| Method | Gibbs | $\langle E \rangle$ |
|---|---|---|
| Rand Init | .2035 | .1026 |
| GraphCL | .1941 | .0995 |
| 3D Infomax QM9 | .1852 | .0968 |
| 3D Infomax Drugs | **.1811** | **.0952** |
| 3D Infomax QMugs | .1835 | .0965 |

While 3D Infomax yields large improvements, the MAE is still substantially higher than that of SMP, which uses the 3D information explicitly. One reason for this is likely that QM9's properties are conformer-specific. There might be a maximum accuracy that can be achieved if only the molecule is known and not for which conformer the property should be predicted. Nevertheless, this performance gap suggests that there is still room for improvement.

Table 2 further confirms that 3D Infomax substantially improves quantum property predictions and generalizes out-of-distribution. Our method outperforms GraphCL, even though GraphCL also sees the fine-tuning molecules during pre-training. Moreover, we observe strong generalization when pre-training with QM9 and fine-tuning on GEOM-Drugs. In this case, the pre-training data only contains the elements C, H, N, O, and F while the target data contains eleven additional elements that are unseen during pre-training.

Such consistent and out-of-distribution improvements can be explained by the type of information captured with 3D Infomax. Learning to reason about molecular geometry and its impact does not depend on the data's molecular space and therefore it is not necessary to have a high similarity between the molecules during pre-training and fine-tuning.

Another advantage of 3D Infomax is its comparably fast convergence. Pre-training on 620k molecules of QMugs with 3 conformers takes 12 hours, compared to 71 hours for GraphCL on 280k molecules of GEOM-Drugs.

## 5.2 PREDICTIVE 3D PRE-TRAINING

Table 3: Comparison of **3D Infomax** against predictive 3D pre-training baselines. Shown is the MAE for predicting QM9's properties. Colors indicate improvement (lower MAE) or worse performance compared to the randomly initialized (**Rand Init**) model.

| Method | $\mu$ | $\alpha$ | homo | lumo | gap | r2 | ZPVE | $c_v$ |
|---|---|---|---|---|---|---|---|---|
| Rand Init | 0.4148 | 0.3348 | 82.10 | 87.74 | 120.94 | 22.14 | 15.08 | 0.1670 |
| Dist-pred | 0.4626 | 0.3570 | 80.58 | 84.93 | 116.21 | 29.23 | 25.91 | 0.1587 |
| Conf-gen | 0.3940 | 0.4219 | 79.75 | 79.16 | 110.72 | 20.86 | 21.10 | 0.1555 |
| 3D Infomax | 0.3512 | 0.2959 | 70.78 | 71.38 | 102.59 | 18.96 | 9.677 | 0.1409 |

3D pre-training by directly predicting 3D quantities is simpler than 3D Infomax and would be preferable in case of similar gains. Therefore, we compare with the baselines in Section 4.2 using the same 140k molecules of GEOM-Drugs for all 3D pre-training methods. *Dist-pred* refers to predicting all atom distances of the highest probability conformer and *Conf-gen* means pre-training by predicting up to 10 conformers. Table 3 shows that 3D Infomax pre-training is always superior to the predictive baselines and is the only method to not suffer from negative transfer (Pan & Yang, 2010).

## 5.3 NON-QUANTUM PROPERTIES

Table 4: Comparison of 3D pre-training baselines and **GraphCL** against **3D Infomax** on various OGB datasets. Shown is either the Root Mean Squared Error (RMSE) (lower is better) or the area under the ROC-curve (ROC-AUC) (higher is better). Colors indicate improvement, worse performance, or no significant change compared to the randomly initialized (**Rand Init**) model.

| Dataset | Metric | Rand Init | Dist-pred | Conf-gen | GraphCL | 3D Infomax |
|---|---|---|---|---|---|---|
| esol | RMSE ↓ | $0.947_{\pm 0.038}$ | $0.986_{\pm 0.025}$ | $0.867_{\pm 0.045}$ | $0.959_{\pm 0.047}$ | $0.894_{\pm 0.028}$ |
| lipo | RMSE ↓ | $0.739_{\pm 0.009}$ | $0.718_{\pm 0.021}$ | $0.757_{\pm 0.035}$ | $0.714_{\pm 0.011}$ | $0.695_{\pm 0.012}$ |
| freesolv | RMSE ↓ | $2.233_{\pm 0.261}$ | $2.486_{\pm 0.222}$ | $2.428_{\pm 0.155}$ | $3.744_{\pm 0.292}$ | $2.337_{\pm 0.227}$ |
| bace | ROC-AUC ↑ | $78.13_{\pm 1.30}$ | $76.51_{\pm 1.95}$ | $80.02_{\pm 1.58}$ | $77.18_{\pm 4.01}$ | $79.42_{\pm 1.94}$ |
| bbbp | ROC-AUC ↑ | $68.27_{\pm 1.98}$ | $66.06_{\pm 1.84}$ | $66.16_{\pm 2.24}$ | $71.06_{\pm 2.00}$ | $69.10_{\pm 1.07}$ |
| tox21 | ROC-AUC ↑ | $73.88_{\pm 0.64}$ | $73.87_{\pm 0.43}$ | $75.24_{\pm 1.00}$ | $78.92_{\pm 0.61}$ | $74.46_{\pm 0.74}$ |
| toxcast | ROC-AUC ↑ | $63.62_{\pm 0.48}$ | $61.58_{\pm 0.58}$ | $64.74_{\pm 1.20}$ | $64.95_{\pm 0.31}$ | $64.41_{\pm 0.88}$ |
| clintox | ROC-AUC ↑ | $58.98_{\pm 5.40}$ | $55.77_{\pm 5.86}$ | $64.27_{\pm 5.22}$ | $51.07_{\pm 5.52}$ | $59.43_{\pm 3.21}$ |
| sider | ROC-AUC ↑ | $55.95_{\pm 3.27}$ | $57.13_{\pm 1.89}$ | $56.34_{\pm 4.20}$ | $57.32_{\pm 5.00}$ | $53.37_{\pm 3.34}$ |
| hiv | ROC-AUC ↑ | $77.06_{\pm 3.16}$ | $75.66_{\pm 1.26}$ | $76.57_{\pm 1.39}$ | $76.06_{\pm 1.06}$ | $76.08_{\pm 1.33}$ |

In the previous sections, we found that 3D Infomax yields large improvements for predicting quantum properties. For non-quantum properties, there is less empirical evidence that 3D information improves prediction accuracy. Nevertheless, for tasks such as binding prediction in the *bace* dataset, we would expect it to be helpful, and we compare different methods pre-trained with GEOM-Drugs.

In Table 4, we find that 3D Infomax improved performance for 4 out of 10 OGB datasets. In contrast to the results for quantum mechanical property predictions (Section 5.1), it is not always superior to

GraphCL and Conf-gen. However, 3D Infomax never decreases performance which can be valuable in practice and make the method worth employing for non-quantum properties as well.

When investigating for which tasks 3D Infomax is useful, we see that abstract tasks such as predicting clinical test outcomes (*clintox*) benefit less. The most significant improvements are rather possible for tasks like predicting solubility and lipophilicity in *esol* and *lipo*. These are more directly related to molecular mechanics and a molecule's intrinsic properties (e.g., the dipole moment/polarity is important for predicting lipophilicity). They do not depend on how a molecule will interact with others to result in, e.g., different effects on patients. For such tasks, *Conf-gen* often leads to significant improvements, providing further evidence for the value of 3D pre-training.

Additionally, for datasets like *bace* with its binding prediction task where 3D information should be valuable, the improvements are only modest. This could suggest that our method does not capture all of the 3D information that is relevant for predicting protein binding, and there is still room for improvement. Another explanation is that a molecule's geometry is less helpful for *bace* since the geometry of the protein and the binding pocket the molecule has to fit into are not known.

## 5.4 NUMBER OF CONFORMERS AND PRE-TRAINING MOLECULES

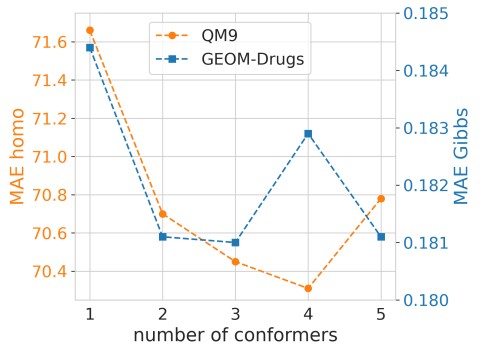 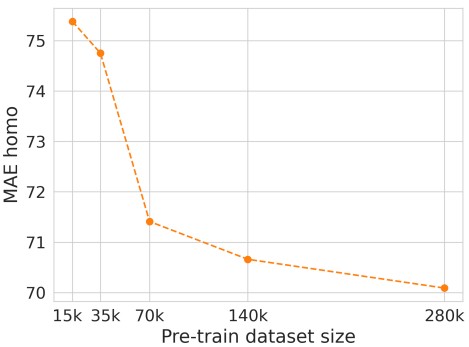

Figure 4: The **left plot** shows the MAE for the QM9's homo and GEOM-Drugs' Gibbs property when varying the number of GEOM-Drugs' conformers used during pre-training. The **right plot** depicts the MAE when using different numbers of molecules of GEOM-Drugs during pre-training.

Figure 4's left plot highlights the benefit of using more than a single conformer. However, the marginal gain reduces as higher energy conformers are added and beyond a certain point (around three conformers), the reduced focus on the most likely conformers worsens the downstream performance. This is in line with the observation that, on average, three conformers are enough to cover 70% of the cumulative Boltzmann weight for GEOM-Drugs. Additionally, experiments in Appendix C.2 show that using multiple conformers is essential when pre-training with QMugs: the MAE for QM9's homo property is $82.57$ with a single conformer while it improves to $70.77$ when using three.

In the right plot of Figure 4 we can observe the performance improving as the size of the pre-training dataset increases. However, the returns are diminishing, and we cannot claim that even larger pre-training datasets are likely to drastically improve performance.

## 6 CONCLUSION

We presented a pre-training strategy, 3D Infomax, that teaches a GNN to produce latent 3D and quantum information from 2D molecular graphs. This can later be used during fine-tuning to improve molecular property predictions while retaining the inference speed of a standard GNN operating on 2D molecular graphs. We found consistently large improvements (∼22%) for quantum properties, overshadowing the gains possible with conventional SSL methods. The embedded 3D knowledge can be transferred across highly different types of molecules (e.g., from molecules with an average of 18 atoms to drug-like molecules with 44.4 atoms) since the representations capture a principled form of information that is known to be useful for molecular tasks. Similarly, we demonstrated that learned embeddings are transferable across different physical, biological, and pharmaceutical tasks. Lastly, we observed that using multiple molecular conformers during pre-training provides valuable additional information to further improve downstream property predictions.

## 7 ETHICS STATEMENT

We see the most important application and the largest possible positive societal impact of our method in the domain of drug discovery. Our 3D Infomax pre-training method improves molecular property predictions with implicit 3D information while being fast at inference. This means that our approach can be used when predicting the properties of vast amounts of molecules to identify potential drug candidates and in large-scale virtual screening. Applications to materials science could also help in developing new materials for energy storage or energy generation (e.g., solar panels), which are crucial for transitioning to clean energy and preventing climate change.

We believe the greatest possible risks to be side effects such as practitioners blindly trusting molecular property predictions of learned models. Research on explainability of GNN's predictions would be crucial for solving this problem. Additional possible risks come from malicious applications of molecular property predictions, such as developing chemical or biological weapons.

## 8 REPRODUCIBILITY STATEMENT

All code to reproduce our results is available at `https://anonymous.4open.science/r/ 3141`. This repository also provides an explanation for running the experiments. Moreover, we detail our hyperparameter search spaces and final parameter settings in Appendix B.3 for all our used architectures and baselines. Furthermore, Appendix B.1 provides the exact datasets we use and their availability with guides for downloading them in our repository. Additionally, there are additional clarifying explanations for the exact architecture of the 3D network in Appendix A. Lastly, we state the seeds which we use for the different runs in our repository as well as Appendix B and completely specify our used hard- and software in Appendix B.5.

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

## A  FURTHER EXPLANATIONS

**3D Network Details** The l-th layer of the 3D network takes two sets as input. First, $n^2 - n$ edge representations $\{d_{uv}^l \in \mathbb{R}^{d_d} \mid u, v \in \mathcal{V} \wedge u \neq v\}$ (the edges of a complete graph without self-loops). In the first layer they are given by the encoded distances fed through an initial feed-forward network $U_{init} : \mathbb{R}^{2F+1} \mapsto \mathbb{R}^{d_d}$ which projects them to the hidden dimension of the edges $d_{uv}^0 = U_{init}(\gamma(d_{uv}))$. The second input is a set of $n$ atom representations $\{h_1^l, \ldots h_n^l\}$ with dimensionality $\mathbb{R}^{d_h}$. In the first layer, the atom representations are all set to the same learned vector that is initialized with a standard normal. With $\|$ meaning concatenation, every layer updates the edge and atom representations and iteratively encodes 3D information into them as follows:

$$m_{uv} = U_{edge}([h_u^l \parallel h_v^l \parallel d_{uv}^l]) \tag{5}$$

$$d_{uv}^{l+1} = d_{uv}^l + m_{uv} \tag{6}$$

$$h_u^{l+1} = U_h([h_u \parallel \sum_{\substack{v=1 \\ v \neq u}}^n m_{uv} * \sigma(U_{softedge}(m_{uv})]). \tag{7}$$

The layer is parameterized by three MLPs where the first one updates the edges $U_{edge} : \mathbb{R}^{2d_h+d_d} \mapsto \mathbb{R}^{d_d}$. The second one updates the atom representations $U_h : \mathbb{R}^{d_h+d_d} \mapsto \mathbb{R}^{d_h}$. The third one $U_{softedge} : \mathbb{R}^{d_d} \mapsto \mathbb{R}$ is followed by the logistic sigmoid function to create a value between 0 and 1 that can be seen as a soft edge weight telling us how probable an edge is for each message $m_{uv}$ as it is done by Satorras et al. (2021).

To produce the final 3D representation $z^b$, all atom representations are aggregated by concatenating their mean, their maximum, and their standard deviation and feeding this through a final feed-forward network $U : \mathbb{R}^{3d_h} \mapsto \mathbb{R}^{d_z}$.

## B  EXPERIMENTAL DETAILS

### B.1  DATA DETAILS

We use three datasets containing 3D information for pre-training with diversity in molecule size and the number of molecules, as can be seen in Table 5. The pre-training datasets are:

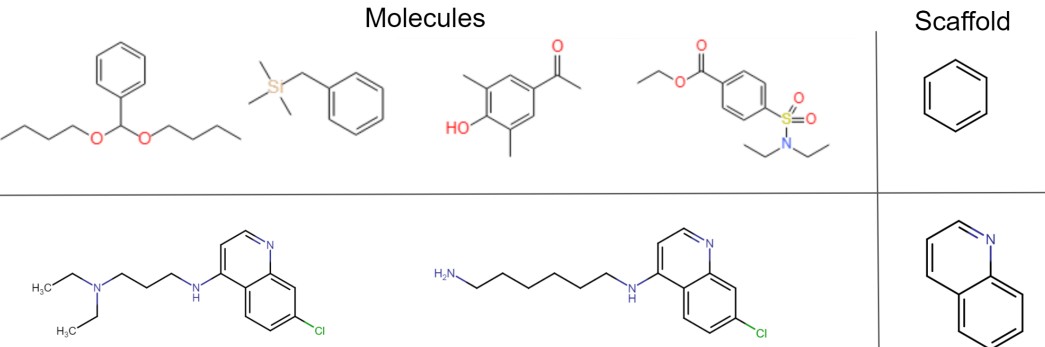

Figure 5: Depiction of two groups of molecules where all the molecules in the top row have the same Bemis-Murcko scaffold (Bemis & Murcko, 1996) which is different from the scaffold to which the two molecules in the bottom row belong. We can easily obtain the scaffold of a molecule using RDKit (Landrum, 2016).

1. **QM9**[1] (Ramakrishnan et al., 2014) contains 134k stable small organic molecules of 5 elements (CHONF). Every molecule has the 3D coordinates of one low-energy conformer and is annotated with 12 quantum mechanical properties as regression targets. The molecules are considered very small, with at most 9 heavy atoms.

2. **GEOM-Drugs**[2] (Axelrod & Gomez-Bombarelli, 2020) consists of 304k realistically-sized biologically and pharmacologically relevant molecules of 16 elements, annotated with multiple 3D conformers, the ensemble Gibbs free energy, and the ensemble energy as regression targets. For the average molecule, 70% of the Boltzmann weight is captured by just three conformers as can be seen in Figure 6a where we also provide a histogram for the number of molecules that have a certain amount of conformers in Figure 6b. The conformers are generated using CREST (Grimme, 2019).

3. **QMugs**[3] (Isert et al., 2021) has 665k drug-like molecules with three diverse conformers each and multiple conformer specific quantum mechanical properties as regression tasks. The conformers are generated using CREST (Grimme, 2019).

For fine-tuning, we use a variety of datasets that cover a wide range of domains and applications. The molecular properties are relevant for quantum mechanics, physical chemistry, biophysics, and physiology such that we can obtain a good estimate of how valuable our 3D pre-training is for each domain. For quantum mechanical properties, which are often specific to a conformer, it is clear that 3D information is important and there has been a lot of evidence that learned methods highly benefit from its use (Klicpera et al., 2020b;a; Liu et al., 2021; Schütt et al., 2017). For these properties, the interest is in how much our method can leverage this information and transfer it to molecules where no 3D geometry is available.

Meanwhile, for biological or physiological properties such as blood-brain barrier penetration, it is not as clear if improvements from 3D information are to be expected. As such, this question needs to be answered next to how much of the benefits 3D pre-training recovers. For this purpose, we use the following molecular graph datasets, which are mainly taken from MoleculeNet (Wu et al., 2017) and we use the scaffold splits[4] with an 80/10/10 split ratio provided by OGB Hu et al. (2020a). The fine-tuning datasets are:

1. **QM9 and GEOM-Drugs:** On these 3D datasets we also fine-tune and evaluate the quantum mechanical properties of one half of the datasets with a random split. This is done after either pre-training on another 3D dataset (generalization), or after pre-training on the other half of the same dataset (in distribution).

---

[1] https://github.com/klicperajo/dimenet/blob/master/data/qm9_eV.npz
[2] https://github.com/learningmatter-mit/geom
[3] https://www.research-collection.ethz.ch/handle/20.500.11850/482129
[4] https://ogb.stanford.edu/docs/graphprop

2. **ESOL:** 1128 common organic small molecules with water solubility data (log solubility in mols per liter).
3. **Lipo:** Experimental data for the octanol/water distribution coefficient of 4200 molecules.
4. **FreeSolv:** The hydration free energy of 642 molecules in water.
5. **HIV:** 41k molecules with binary labels for HIV virus replication inhibition.
6. **BACE:** Binary labels of binding results for inhibitors of human $\beta$-secretase 1 for 1512 molecules.
7. **BBBP:** 2039 molecules with binary labels for blood-brain barrier penetration.
8. **Tox21:** 7831 molecules with binary labels of their toxic for 12 different targets.
9. **ToxCast:** 8576 molecules with binary labels of toxicity experiment outcomes with 617 targets.
10. **SIDER:** 1427 approved drugs with 27 different side effect groups and the task is to predict whether the drug is in the side effect group.
11. **ClinTox:** 1477 drugs with two binary annotations where the first is to predict toxicity in clinical trials and the second is the FDA approval status.

The reason why *muv* and *pcba* are the only datasets from the OGB benchmark suite which we omit is their larger size.

Table 5: Statistics of the used datasets. In the upper section are datasets with 3D information, which we use for pre-training, and the datasets in the bottom section do not contain additional 3D annotations.

| Dataset | #Molecules | Avg. #Atoms | Avg. #Bonds | split |
|---|---|---|---|---|
| QM9 | 130 831 | 18.0 | 18.6 | random |
| GEOM-Drugs | 304 293 | 44.4 | 46.4 | random |
| QMugs | 665 911 | 30.6 | 33.4 | random |
| esol | 1128 | 13.3 | 13.7 | scaffold |
| lipo | 4200 | 27.0 | 29.5 | scaffold |
| freesolv | 642 | 8.7 | 8.4 | scaffold |
| bace | 1512 | 34.1 | 36.9 | scaffold |
| bbbp | 2039 | 24.1 | 26.0 | scaffold |
| hiv | 41 127 | 25.5 | 27.5 | scaffold |
| tox21 | 7831 | 18.6 | 19.3 | scaffold |
| toxcast | 8576 | 18.8 | 19.3 | scaffold |
| clintox | 1477 | 26.2 | 27.9 | scaffold |
| sider | 1427 | 33.6 | 35.4 | scaffold |

## B.2 UNITS AND MEANING OF QUANTUM PROPERTIES

For the GEOM-Drugs dataset, all reported numbers have the unit kcal/mol, Gibbs refers to the ensemble Gibbs free energy, and $\langle E \rangle$ to the ensemble energy.

Table 6: Units and description of quantum mechanical properties of the QM9 dataset.

| Property | Unit | Description |
|---|---|---|
| $\mu$ | Debye | Dipole moment |
| $\alpha$ | $Bohr^3$ | Isotropic polarizability |
| $homo$ | meV | Energy of Highest occupied molecular orbital (HOMO) |
| $lumo$ | meV | Energy of Lowest occupied molecular orbital (LUMO) |
| $gap$ | meV | Gap, difference between LUMO and HOMO |
| $r2$ | $Bohr^2$ | Electronic spatial extent |
| $ZPVE$ | meV | Zero point vibrational energy |
| $c_v$ | $\frac{cal}{molK}$ | Heat capacity at 298.15 K |

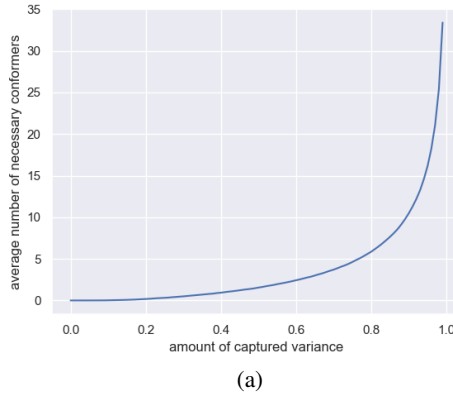 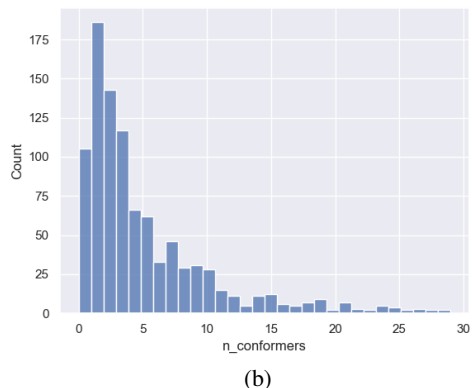

(a)  (b)

Figure 6: a) The average number of conformers necessary to cover a certain amount of Boltzmann weight in GEOM-Drugs. For a given amount of cumulative Boltzmann weight on the horizontal axis, the vertical axis shows the average number of conformers necessary to pass that threshold. b) Histogram of how many molecules there are in GEOM-Drugs with a certain amount of conformers. The histogram is created for 1000 molecules of GEOM-Drugs.

### B.3 PARAMETER DETAILS

The hyperparameters for SMP are taken from the official repository[5] where Liu et al. (2021) provide their code, and we predict $gap$ even though it could be calculated as $|homo - lumo|$. The parameter search space and final parameters for the PNA architecture are specified in Table 7 and those of the 3D network in Table 8.

**Pre-training:** We use Adam with a start learning rate of $8 \times 10^{-5}$ and a batch size of 500. The learning rate schedule during pre-training starts with 700 optimization steps of linear warmup followed by the schedule given by the *ReduceLROnPlateau* scheduler by PyTorch[6] with reduction parameter 0.6, patience 25, and a cooldown of 20.

**Fine-tuning quantum mechanical properties:** We use Adam with a start learning rate of $7 \times 10^{-5}$, weight decay $1 \times 10^{-11}$ and a batch size of 128. For the learning rate schedule, we first perform warmup as follows. We consider three different sets of learnable parameters: (1) the batch norm parameters, (2) all newly initialized parameters that were not transferred, and (3) all parameters. For these sets, we increase the learning rate in this order from 0 to the start learning rate with linear interpolation. For parameter group one, we warm up for 700 steps, 700 steps for group 2, and 350 steps for group 3. After that we use the schedule given by the *ReduceLROnPlateau* with reduction parameter 0.5, patience 25, and a cooldown of 20.

**Fine-tuning non-quantum properties:** We use Adam with a start learning rate of $1 \times 10^{-3}$ and a batch size of 32. The learning rate schedule is the same as for the quantum mechanical properties.

The experiment on the non-quantum properties has different hyperparameters for PNA since the smaller datasets are easily overfitted on with the large architecture we use for the quantum mechanical properties. A smaller PNA yields better performance with the random initialization baseline. Therefore the PNA in these experiments has a hidden dimension of 50 and 3 message passing layers as propagation depth. Apart from that, it is the same as the PNA described in Table 7.

---

[5] https://github.com/divelab/DIG
[6] https://pytorch.org/docs/stable/generated/torch.optim.lr_scheduler.ReduceLROnPlateau.html

Table 7: Search space for the 2D network PNA through which we searched to obtain a strong baseline performance on the energy of the highest occupied molecular orbital (homo) property of the QM9 dataset. The parameters were tuned in the order in which they are listed in this table from top to bottom. After this was completed for all parameters, we performed a second round of tuning for a subset of them. The final parameters are marked in **bold**.

| Parameter | Search Space |
|---|---|
| propagation depth | [4, 5, 6 ,**7**] |
| hidden dimension | [40, 50, 75, 90, 100, 150, **200** ,300] |
| message MLP layers | [1, **2**, 3] |
| update MLP layers | [**1**, 2, 3] |
| aggregators | [mean, max, min, std, sum], [mean, max, min], [mean, max, sum], **[mean, max, min, std]**, [max, sum], [sum] |
| scalers | [identity], **[identity, amplification, attenuation]** |
| readout aggregators | [mean], [sum], [mean, max, sum], **[mean, max, min, sum]** |
| dropout | [**0**, 0.05, 0.1, 0.2] |
| batchnorm after MLPs | **True**/False |
| batchnorm in MLPs | **True**/False |
| readout MLP layers | [1, **2**, 3] |
| batchnorm momentum | [**0.1**, 0.9, 0.93] |

Table 8: Search space for the 3D network *Net3D* through which we searched to obtain a strong baseline performance on the homo property of the QM9 dataset and we considered the size of the network where parameters leading to less memory use are preferred. The parameters were tuned in the order in which they are listed in this table from top to bottom. After this was completed for all parameters, we performed a second round of tuning for a subset of them. The final parameters are marked in **bold**.

| Parameter | Search Space |
|---|---|
| propagation depth | [**1**, 3, 4, 5] |
| hidden dimension | [10, **20**, 40, 60, 80, 100] |
| F used in $\gamma : \mathbb{R} \mapsto \mathbb{R}^{2F+1}$ | [0, 3, **4**, 8, 10, 50] |
| message MLP layers | [**1**, 2, 3] |
| update MLP layers | [**1**, 2, 3] |
| readout aggregators | [mean], [sum], **[mean, max, min]**, [mean, max, min, sum] |
| dropout | [**0**, 0.05, 0.1, 0.2, 0.5] |
| batchnorm after MLPs | **True**/False |
| readout MLP layers | [**1**, 2, 3] |
| batchnorm momentum | [0.1, 0.9, **0.93**] |

## B.4 Confidence Interval Details

All the specified confidence intervals in our work are standard deviations calculated from different weight initializations using the seeds $[1, 2, 3, 4, 5, 6]$ or $[1, 2, 3, 4]$. The following tables provide additional confidence intervals for the results in the main text.

Table 9: Additional confidence intervals of our method in Table 1. All standard deviations are calculated from 4 seeds except for the homo property where 6 are used.

| Target | Rand Init | Our 3D Infomax | | |
|--------|-----------|----------------|-------|-------|
| | | QM9 | Drugs | QMugs |
| $\mu$ | 0.4133±0.003 | **0.3507**±0.005 | **0.3512**±0.010 | 0.3668±0.004 |
| $\alpha$ | 0.3972±0.014 | 0.3268±0.006 | 0.2959±0.009 | **0.2807**±0.012 |
| homo | 82.10±0.33 | **68.96**±0.32 | 70.78±0.82 | 70.77±0.74 |
| lumo | 85.72±1.62 | **69.51**±0.54 | 71.38±0.74 | 78.10±0.69 |
| gap | 123.08±3.98 | **101.71**±2.03 | 102.59±3.27 | 103.85±1.92 |
| r2 | 22.14±0.21 | **17.39**±0.94 | 18.96±0.69 | 18.00±0.40 |
| ZPVE | 15.08±2.83 | 7.966±1.87 | 9.677±1.29 | 12.06±2.40 |
| $c_v$ | 0.1670±0.004 | 0.1306±0.009 | 0.1409±0.016 | **0.1208**±0.008 |

Table 10: Additional confidence intervals for Table 2.

| Method | Gibbs | $\langle E \rangle$ |
|--------|-------|---------------------|
| Rand Init | .2035± 0.0011 | .1026± 0.0017 |
| GraphCL | .1941 | .0995 |
| 3D Infomax QM9 | .1852 | .0968 |
| 3D Infomax Drugs | **.1811** | **.0952** |
| 3D Infomax QMugs | .1835 | .0965 |

Table 11: Additional confidence intervals for Table 3.

| Method | $\mu$ | $\alpha$ | homo | lumo | gap | r2 | ZPVE | $c_v$ |
|--------|-------|----------|------|------|-----|-----|------|-------|
| Rand Init | ±0.003 | ±0.014 | ±0.33 | ±1.62 | ±3.98 | ±0.21 | ±2.83 | ±0.004 |
| 3D Infomax | ±0.010 | ±0.009 | ±0.82 | ±0.74 | ±3.27 | ±0.69 | ±1.29 | ±0.016 |

## B.5 Implementation

Code to 3D pre-train a GNN or to reproduce results is available at `https://anonymous.4open.science/r/3141`. All experiments were implemented in *PyTorch* (Paszke et al., 2017) using the deep learning libraries for processing graphs *Pytorch Geometric* (Fey & Lenssen, 2019) and *Deep Graph Library* (Wang et al., 2019). The code we use for SMP (Liu et al., 2021) is under the GNU General Public License v3.0 and we use their implementation after discussing it with the first author of the paper and under the consideration that their project welcomed our contributions to their library.

The experiments were conducted on two different machines while the same system was always used in direct comparisons. The first machine has an AMD Ryzen 1700 CPU @ 3.70Ghz, 16GB of RAM, and an Nvidia GTX 1060 GPU with 6GB vRAM. The second system contains two Intel Xeon Gold 6248 CPUs @ 2.50GHz each with 20/40 cores, 400GB of RAM, and four Quadro RTX 8000 GPUs with 46GB vRAM of which only a single one was used for each experiment. All mentions and of training time refer to the second system.

Table 12: Comparison of 3D networks. The MAE of the homo property pre-training and fine-tuning on different halves of QM9. *Net3D w/o* $\gamma$ refers to dropping the distance encoding of Net3D. *Net3D* achieves the best MAE.

| Method | QM9 MAE |
|---|---|
| Rand Init | $82.10_{\pm 0.33}$ |
| SMP | 72.37 |
| EGNN | 70.46 |
| *Net3D* w/o $\gamma$ | 70.34 |
| *Net3D* | $\mathbf{68.96_{\pm 0.32}}$ |

## C  ADDITIONAL RESULTS

### C.1  DIFFERENT 3D NETWORKS AND ABLATION

In this section, we justify the design of our 3D network, which we call *Net3D* in this comparison. In Table 12 we compare *Net3D* with different alternative 3D networks, which are the 3D GNNs SMP and EGNN operating on learned node embeddings similar to *Net3D*. Additionally, we ablate the use of our $\gamma$ function that maps the pairwise distances to a higher dimensional space since it would be an unnecessary complication if it provides no benefit. We call it *Net3D w/o* $\gamma$ if *Net3D* directly operates on the pairwise distances.

In Table 12 we can observe that *Net3D* yields the best downstream performance and that using our $\gamma$ function is a valuable component of it. Possibly EGNN would benefit similarly from this encoding. SMP's downstream performance is the worst which could be expected since the 3D input representation which it uses does not uniquely define all the relative positions in a molecule.

We note that SMP is able to distinguish chiral molecules, unlike the other 3D networks, but this advantage cannot be evaluated with our experiments on quantum mechanical properties. Chirality only becomes relevant when considering the interactions between molecules, and in these situations, SMP might be able to leverage its advantage such that our evaluation could be criticized as not entirely fair. Additionally, SMP has much lower memory requirements since it does not suffer from the quadratic complexity of EGNN and *Net3D* in the molecule size. Nevertheless, *Net3D* performs the best, and for drug-like molecules, the quadratic complexity is not problematic.

### C.2  DIFFERENT METHODS FOR MULTIPLE CONFORMERS

We test three main approaches and variations of them for incorporating the 3D information of multiple conformers to justify our choice in the main text. The most straightforward one is **conformer sampling**. We use one of the single conformer setups, but when sampling the batch, we additionally sample $j \in \{1 \dots c_i\}$ and use the single conformer $R_i^j$. The probability of sampling a conformer is either distributed uniformly (so $1/c_i$ is the probability for each $j$) or given by the Boltzmann weight of each conformer.

**multi3D** is the approach from the main text where we include multiple conformers as additional positive pairs in contrastive learning. For each molecule $(G_i, \{R_i^j\}_{j \in \{1 \dots c_i\}})$ we choose the $c$ lowest energy conformers to have a fixed number of them. If there are fewer than $c$ conformers for a molecule ($c_i < c$), then the lowest energy conformer is repeated. For every molecule the 3D network now takes all $c$ conformers $\{R_i^j\}_{j \in \{1 \dots c\}}$ as input and produces their latent 3D representations $\{z_{i,j}^b\}_{j \in \{1 \dots c\}}$ which we can see as additional positive samples. In our contrastive setting, we, therefore, want the similarity between $z_i^a$ and all conformer representations that come from the same molecule $z_{i,j}^b$ to be high. As such, we modify our loss to obtain:

$$\mathcal{L}^{multi3D} = -\frac{1}{N} \sum_{i=1}^{N} \left[ log \frac{\sum_{j=1}^{c} e^{sim_{cos}(z_i^a, z_{i,j}^b)/\tau}}{\sum_{\substack{k=1 \\ k \neq i}}^{N} \sum_{j=1}^{c} e^{sim_{cos}(z_i^a, z_{k,j}^b)/\tau}} \right]. \tag{8}$$

One concern with this formulation is the following. Let us consider a single molecule. The objective of high similarity between the many 3D representations and the single 2D representation might be easier to solve through encoding the same 2D information in the 3D representations instead of capturing the 3D information of all conformers in the single 2D representation. The 2D network would therefore not learn to produce 3D information from its 2D inputs because the mutual information could be maximized through encoded 2D information.

To address this problem, **multi3D+2D** is our third approach. The 2D network is now modified to produce $c$ many latent 2D representations $f^a(G_i) = \{\boldsymbol{z}^a_{i,j}\}_{j \in \{1...c\}}$ which are compared to all 3D representations of the same molecule in a similarity function $sim$. We simply use this similarity in the loss instead of the cosine similarity. Intuitively, the 2D network now has to produce an embedding for each 3D conformer.

One way to define such a similarity between two same-sized sets of vectors is to use the sum of all pairwise cosine similarities (for brevity we drop the subscript and only write $\{\boldsymbol{z}^a_{i,j}\}$ to mean the set of all representations corresponding to the i-th molecule):

$$sim_{all}(\{\boldsymbol{z}^a_{i,j}\}, \{\boldsymbol{z}^b_{i,j}\}) = \sum_{j=1}^{c} \sum_{k=1}^{c} sim_{cos}(\boldsymbol{z}^a_{i,j}, \boldsymbol{z}^b_{i,k}) \tag{9}$$

More principled would be to find the optimal transport matching with the highest cosine similarity, such that one 2D representation always corresponds to one 3D representation. However, this approach was not computationally feasible with the batch sizes we use in contrastive learning. We instead opt for an upper bound on the maximum similarity matching. For every 3D representation, we choose the 2D representation that has the highest similarity. This way, one 2D representation could be associated with multiple 3D embeddings, and we no longer have a mass preserving matching:

$$sim_{max}(\{\boldsymbol{z}^a_{i,j}\}, \{\boldsymbol{z}^b_{i,j}\}) = \sum_{k=1}^{c} \max_{j \in \{1...c\}} sim_{cos}(\boldsymbol{z}^a_{i,j}, \boldsymbol{z}^b_{i,k}). \tag{10}$$

Beyond these similarity measures, we explore additional ones based on the inverse of different distance functions and asymmetric metrics such as the maximum mean discrepancy (Gretton et al., 2012) or the KL- and JS-Divergence when interpreting the conformer representations as samples from a normal distribution.

**Results** We evaluate which of the different approaches best leverage the additional conformer's information to justify our choice for *multi3D* in the main text. Another hypothesis we wish to test is that for smaller molecules such as those in QM9, the ability to make predictions informed by multiple conformers is not as important as for larger drug-like molecules. The reasoning is that a single conformer takes most of the Boltzmann weight for QM9's molecules due to the fewer degrees of freedom.

We test *conformer sampling*, *multi3D*, and *multi3D+2D* when pre-training on either QMugs or one half of GEOM-Drugs and fine-tuning on QM9 or the other half of GEOM-Drugs. In QMugs we have three diverse conformers available for each molecule which are all used, while for GEOM-Drugs different numbers of conformers are available of which we use the five with the highest Boltzmann weight i.e. lowest energy. If there are fewer than five we duplicate the lowest energy conformer (see Section 4.1 for details). We recall that for the *multi3D+2D* loss sets with as many elements as conformers are produced by the 2D and 3D networks. Both the discussed $sim_{all}$ and $sim_{max}$ are used as similarity measures between those sets. For the *conformer sampling* strategies of using a uniform weighting or sampling conformers according to their Boltzmann weight, we do not evaluate the latter on QMugs since we do not have it available with exactly three conformers per molecule.

In Table 13 we can observe that there are large improvements possible when using multiple conformers. After pre-training on QMugs, the MAE, when predicting the homo property, decreases from 82.57 to 70.77 and from .1966 to .1831 for predicting the Gibbs free energy for GEOM-Drugs. Notably, these improvements are much larger than when pre-training with GEOM-Drugs. This is likely because the GEOM-Drugs dataset contains the lowest energy conformers, and we always use the most probable one with the highest Boltzmann weight when pre-training with a single conformer. Meanwhile, the QMugs dataset contains three diverse conformers per molecule and not

Table 13: Comparison of strategies for using multiple conformers. The middle double-column shows the results for pre-training on one half of GEOM-Drugs and the right double-column corresponds to pre-training on QMugs, and the second row indicates what dataset was used for fine-tuning. The ***Random Init*** row shows the performance when training from scratch without any pre-training. For QM9, the reported number is the MAE of the homo property, and for GEOM-Drugs it is the MAE when predicting the ensemble Gibbs free energy. There are large improvements from using multiple conformers, but the differences between the methods are small.

| Loss/Estimator | `GEOM-Drugs` pre-training | | `QMugs` pre-training | |
| --- | --- | --- | --- | --- |
| | `QM9` | `GEOM-Drugs` | `QM9` | `GEOM-Drugs` |
| Rand Init | $82.10_{\pm 0.33}$ | $.2035_{\pm .0011}$ | $82.10_{\pm 0.33}$ | $.2035_{\pm .0011}$ |
| single conformer | 71.66 | .1844 | 82.57 | .1966 |
| uniform sampling | **70.66** | **.1823** | 72.94 | .1874 |
| boltzmann sampling | **70.93** | .1846 | x | x |
| multi3D | **70.78** | **.1811** | **70.77** | **.1831** |
| $sim_{all}$ | 71.11 | .1849 | 72.40 | .1936 |
| $sim_{max}$ | **70.81** | .1896 | **71.15** | **.1840** |

the ones with the highest Boltzmann weight. Pre-training with the lowest energy conformer from GEOM-Drugs already captures most of the relevant information, and using more is not as beneficial. However, for QMugs, using the information of all three diverse conformers is crucial.

Similar to the small improvements over the random initialization baseline with GEOM-Drugs, the different methods for using multiple conformers mostly perform the same when pre-training with GEOM-Drugs. When pre-training with QMugs instead, the MAEs are overall slightly worse, and we find *multi3D* to perform the best. Note that this is with the slight caveat that the epoch at which pre-training is stopped for all methods was chosen based on where *multi3D* had the lowest MAE.

Due to these results, we consider *multi3D*, and *conformer sampling* with uniform weighting as our best methods since *multi3D* performs slightly better with pre-training on QMugs but *conformer sampling* is simpler and especially uses much less memory. For *multi3D*, all the conformers need to be processed in parallel, and training with more than 5 conformers and a batch size of 500 would not be possible on a 48GB vRAM GPU.

The hypothesis that the downstream performance on the smaller molecules of QM9 would benefit less from using multiple conformers than the molecules of GEOM-Drugs clearly does not hold. Surprisingly, the improvements on the small molecules of QM9 are larger.

## C.3 DIFFERENT LOSSES

We compare the different losses to estimate and maximize the mutual information. For this purpose, we pre-train PNA on $50\,000$ molecules from QM9 and another instance on $140\,000$ molecules of GEOM-Drugs, both with a single conformer. We do so with the Donsker-Varadhan (Hjelm et al., 2019) estimator, the Jensen-Shannon estimator (Hjelm et al., 2019), noise contrastive estimation of mutual information (InfoNCE), and our loss. For our loss, we search over seven temperature parameters $\tau \in [0.05, 0.1, 0.2, 0.3, 0.4, 0.5, 0.7]$ and choose $\tau = 0.1$.

In Table 14 we see that 3D pre-training on GEOM-Drugs or QM9 can yield significant improvements for predicting quantum mechanical properties, especially when using InfoNCE and our loss as objectives. These two objectives perform better than the Donsker-Varadhan, and Jensen-Shannon estimator in every case and the Jensen-Shannon objective is superior to the Donsker-Varadhan estimator, which seems to yield no significant improvements over random initialization. The superiority of the Jensen-Shannon loss over the Donsker-Varadhan alternative is in line with the findings of Hjelm et al. (2019) in their different setting on images. While our loss seems to perform better than InfoNCE in three settings, this might be due to the additional investment in searching through temperature parameters for our loss.

Table 14: Comparison of mutual information estimators for 3D Infomax. The middle double-column shows the results for pre-training on one half of QM9, and the right double-column corresponds to pre-training on one half of GEOM-Drugs, and the second row indicates what dataset was used for fine-tuning. The **_Rand Init_** row shows the performance when training from scratch without any pre-training. For QM9 the reported number is the MAE of the homo, and for GEOM-Drugs it is the MAE when predicting the ensemble Gibbs free energy.

| Loss/Estimator | QM9 pre-training | | GEOM-Drugs pre-training | |
|---|---|---|---|---|
| | QM9 | GEOM-Drugs | QM9 | GEOM-Drugs |
| Rand Init | $82.10_{\pm0.33}$ | $.2035_{\pm.0011}$ | $82.10_{\pm0.33}$ | $.2035_{\pm.0011}$ |
| Donsker-Varadhan | 82.49 | .2152 | 85.46 | .2013 |
| Jensen-Shannon | 80.71 | .2078 | 81.61 | .2047 |
| InfoNCE | 75.81 | **.1938** | 79.31 | .1894 |
| our loss | **$68.96_{\pm0.32}$** | .1945 | **71.66** | **.1844** |

## C.4 SSL METHODS

Here we compare our 3D Infomax pre-training against three additional SSL methods. These are Barlow Twins (Zbontar et al., 2021), multi-modal Bootstrap your own latent (BYOL) (Grill et al., 2020), and Variance-Invariance-Covariance Regularization (VICReg) (Bardes et al., 2021). We pre-train these methods on one half of QM9. For a fair comparison, we search through 8 different hyperparameter settings based on the downstream performance on the QM9 homo property. After these method-specific hyperparameters were selected, we tuned every method with a random search over the same search space.

For 3D Infomax, we vary the temperature of our loss $\tau$. When using BYOL we try different decay rates $\gamma$ for the exponential moving average weight copying. Here, we include $\gamma = 0$ making our setup similar to a multi-modal version of SimSiam (Chen & He, 2020). For Barlow Twins, the hyperparameter is $\lambda$ weighting the redundancy loss. Lastly, for VICReg we vary $\mu$ and $\nu$, the parameters for the variance and the covariance regularization:

1. 3D Infomax with our loss: $\tau \in [0.02, 0.05, 0.1, 0.2, 0.3, 0.4, 0.5, 0.7]$ where $\tau = 0.01$ performed the best.

2. Multi-modal BYOL: $\gamma \in [0, 0.0005, 0.001, 0.005, 0.01, 0.03, 0.05, 0.07]$ where $\gamma = 0.005$ performed the best.

3. Barlow-Twins: $\lambda \in [0.002, 0.0039, 0.005, 0.007, 0.01, 0.012, 0.015, 0.02]$ where $\gamma = 0.0039$ performed the best.

4. VICReg: $\lambda = 1$; $\mu \in [1, 0.5]$; $\nu \in [0.02, 0.04, 0.1, 0.3]$ where $\lambda = 1, \mu = 1, \nu = 0.04$ performed the best

Table 15: Comparison of latent space SSL methods. The numbers show the MAE when predicting QM9's homo property after pre-training on one half of QM9 with the given method and fine-tuning on the other half of QM9. The **_Rand Init_** column shows the MAE without pre-training and with random weight initialization. 3D Infomax is our best latent space SSL method.

| | Random Init | 3D Infomax | BYOL | Barlow Twins | VICReg |
|---|---|---|---|---|---|
| QM9 MAE | $82.10_{\pm0.33}$ | **$68.96_{\pm0.32}$** | $79.16_{\pm0.58}$ | $82.38_{\pm0.48}$ | 85.15 |

The results in Table 15 showcase that 3D Infomax clearly is the superior method in our setting. It decreases the MAE from $82.10 \pm 0.33$ to $68.96 \pm 0.32$ while the other methods either lead to no improvement or to the much smaller drop to $79.16 \pm 0.58$ for BYOL. This is not due to collapse to a constant solution since we can observe a high variance between the representations in a batch for all methods. Furthermore, with the final parameter settings, all methods were able to achieve a low value for their loss during pre-training, both on the training and validation data and there are no optimization issues.

Intuitively, the results can be explained by 3D Infomax being the only method that optimizes a lower bound on the mutual information, which potentially makes it especially fit for our setting. The other approaches have no direct relation to mutual information and instead rely on maximizing a notion of similarity with tricks to prevent collapse. While this might work for conventional SSL, we see no success in our scenario where the rigorous guarantee on maximizing the mutual information seems valuable.

Another reason for the poor performance of BYOL and especially Barlow Twins and VICReg might be that they rely on having symmetric networks to generate the compared representations. In our scenario, we have very little similarity between the architectures with our 2D and 3D networks operating on different modalities. This hypothesis would fit in line with the findings of Bardes et al. (2021) and Zbontar et al. (2021) where introducing asymmetries between the networks hurt performance.

## C.5 PRE-TRAINING A 3D GNN

We try to use our 3D Infomax setup to pre-train a 3D GNN. For this purpose, we employ SMP (Liu et al., 2021) as 3D network during pre-training with half of the QM9 dataset. We then transfer it's weights and fine tune them using the accurate 3D conformers of the other half of QM9's molecules to predict the dataset's properties. We compare this with SMP trained on the same molecules with randomly initialized weights. The only architectural difference between the networks is that the pre-trained GNN does not use atom features for the reasons explained in the 3D Network paragraph in Section 4.1.

Table 16: MAE for predicting QM9's molecular properties. SMP is tested with random weight initialization and with the weights obtained from using it as 3D network in our 3D Infomax pre-training setup.

| Target | SMP Rand Init | SMP pre-trained |
|--------|---------------|-----------------|
| $\mu$ | 0.0726 | 0.0801 |
| $\alpha$ | 0.1542 | 0.1276 |
| homo | 56.19 | 44.50 |
| lumo | 43.58 | 37.48 |
| gap | 85.10 | 70.45 |
| r2 | 1.51 | 1.12 |
| ZPVE | 2.69 | 2.43 |
| $c_v$ | 0.0498 | 0.0421 |

In Table 16, we find that pre-training improves the 3D GNN's performance. This may be due to the covalent bonding structure and other 2D edge information that is available during pre-training and which SMP usually cannot use since it employs a radius graph. This is the case even though the pre-trained SMP does not have access to the atom features. Pre-training 3D GNNs might be an interesting future direction to attempt beating the state-of-the-art methods for predicting quantum properties with accurate 3D information.

## C.6 CHEAP NEURAL CONFORMERS AS 3D GNN INPUT

In Section 5.1 we used RDKit's ETKDG algorithm (Landrum, 2016) to generate inaccurate but cheap and fast conformers and employed them as inputs to the 3D GNN SMP (Liu et al., 2021). Here, we attempt the same with conformers generated by the SOTA deep learning method for conformation generation which is GeoMol (Ganea et al., 2021). For this purpose, we train GeoMol with 50k molecules of QM9 and use it to generate the conformations for the rest of QM9. SMP is then trained on 50k different molecules to predict their properties, either using RDKit's conformers or those of GeoMol. This enables a fair comparison with 3D Infomax, which uses the same molecules for pre-training that were used to train GeoMol. When visually inspecting some of the conformers generated by GeoMol, we found that they were sometimes of poor quality for molecules with rings and contained outliers with conformations that seem particularly unrealistic.

Table 17: MAE for QM9's properties. **3D Infomax** is tested with three different pre-training datasets and compared with the 3D GNN SMP using explicit 3D coordinates. The conformers are generated using either the classical method RDKit ETKDG or the learned method GeoMol. Colors indicate improvement (lower MAE) or worse performance compared to the randomly initialized (**Rand Init**) model.

| Target | Rand Init | Our 3D Infomax | | | RDKit SMP | GeoMol SMP |
|---|---|---|---|---|---|---|
| | | QM9 | Drugs | QMugs | | |
| $\mu$ | $0.4133_{\pm 0.003}$ | **0.3507** | **0.3512** | 0.3668 | 0.4344 | 0.6046 |
| $\alpha$ | $0.3972_{\pm 0.014}$ | 0.3268 | 0.2959 | **0.2807** | 0.3020 | 0.8435 |
| homo | $82.10_{\pm 0.33}$ | **68.96** | 70.78 | 70.77 | 82.51 | 195.0 |
| lumo | $85.72_{\pm 1.62}$ | **69.51** | 71.38 | 78.10 | 80.36 | 201.4 |
| gap | $123.08_{\pm 3.98}$ | **101.71** | 102.59 | 103.85 | 114.24 | 284.1 |
| r2 | $22.14_{\pm 0.21}$ | **17.39** | 18.96 | 18.00 | 22.63 | 65.84 |
| ZPVE | $15.08_{\pm 2.83}$ | 7.966 | 9.677 | 12.06 | **5.18** | 17.40 |
| $c_v$ | $0.1670_{\pm 0.004}$ | 0.1306 | 0.1409 | **0.1208** | 0.1419 | 0.5467 |

Table 17 shows that SMP performs poorly with the conformers generated by GeoMol and using those generated by RDKit is always superior. This is the case even though the average accuracy of GeoMol's conformers is comparable to that of RDKit ETKDG's conformers when GeoMol is trained on all of QM9 (Ganea et al., 2021). We hypothesize that the high MAEs with GeoMol's conformers occur since they contained some particularly unrealistic outlier conformations, and SMP is not able to handle those well.

## C.7 COMBINING PRE-TRAINING METHODS

Here we simply use GraphCL's node drop augmentation for the 2D graph and the 3D information (removing all pairwise distances for a removed atom) with a drop ratio of 0.2 during our 3D pre-training process.

Table 18: Comparison of performance when combining 3D pre-training with conventional pre-training by randomly dropping nodes on the 2D or 3D side (labeled **3D Infomax +** ) for various biophysical property OGB datasets. **GraphCL** is another pre-trained baseline. Shown is either the RMSE indicated by ↓ where lower values are better or the area under the curve of the Receiver Operator Characteristic (ROC-AUC) indicated by ↑ where higher values are better. Colors indicate improvement, worse performance, or no significant change compared to the randomly initialized (**Rand Init**) model. 3D Infomax is either on par with random initialization or better. There is no negative transfer as there is with GraphCL.

| dataset | Rand Init | GraphCL | 3D Infomax | 3D Infomax + |
|---|---|---|---|---|
| esol↓ | $0.947_{\pm 0.038}$ | $0.959_{\pm 0.047}$ | $0.894_{\pm 0.028}$ | $0.918_{\pm 0.037}$ |
| lipo↓ | $0.739_{\pm 0.009}$ | $0.714_{\pm 0.011}$ | $0.695_{\pm 0.012}$ | $0.710_{\pm 0.007}$ |
| freesolv↓ | $2.233_{\pm 0.261}$ | $3.744_{\pm 0.292}$ | $2.337_{\pm 0.227}$ | $2.791_{\pm 0.323}$ |
| bace↑ | $78.13_{\pm 1.30}$ | $77.18_{\pm 4.01}$ | $79.42_{\pm 1.94}$ | $79.28_{\pm 3.61}$ |
| bbbp↑ | $68.27_{\pm 1.98}$ | $71.06_{\pm 2.00}$ | $69.10_{\pm 1.07}$ | $68.64_{\pm 2.19}$ |
| tox21↑ | $73.88_{\pm 0.64}$ | $78.92_{\pm 0.61}$ | $74.46_{\pm 0.74}$ | $73.73_{\pm 0.69}$ |
| toxcast↑ | $63.62_{\pm 0.48}$ | $64.95_{\pm 0.31}$ | $64.41_{\pm 0.88}$ | $63.95_{\pm 0.38}$ |
| clintox↑ | $58.98_{\pm 5.40}$ | $51.07_{\pm 5.52}$ | $59.43_{\pm 3.21}$ | $83.59_{\pm 3.64}$ |
| sider↑ | $55.95_{\pm 3.27}$ | $57.32_{\pm 5.00}$ | $53.37_{\pm 3.34}$ | $58.43_{\pm 1.28}$ |
| hiv↑ | $77.06_{\pm 3.16}$ | $76.06_{\pm 1.06}$ | $76.08_{\pm 1.33}$ | $75.38_{\pm 0.95}$ |

