# OpenReview forum: "3D Pre-training improves GNNs for Molecular Property Prediction"
_ICLR.cc/2022/Conference — ICLR 2022 Submitted_

### Official Review · Reviewer_7nK8 · 2021-10-18

**Correctness:** 3
**Technical Novelty And Significance:** 3
**Empirical Novelty And Significance:** 3
**Recommendation:** 5
**Confidence:** 4

**Main Review:**

Strengths:

1. The method proposed in this paper is novel and sound. Incorporating 3D information for molecular property prediction should improve its performance as demonstrated by this and previous work.

2. The experiments are well designed. The authors choose to pre-train on three datasets (QM9, GEOM-Drugs, and QMugs) and evaluate for both quantum mechanical properties and non-quantum properties. Yet some setting descriptions and results are missing (see weaknesses).

3. The paper is clearly written and easy to follow (with some exceptions shown below).


Weaknesses:

1. Some experimental settings are unclear. For example, for QM9 and other datasets, what are the input features for your method and the baselines? There is a bunch of work using QM9 for evaluation. If they follow the same setting, the authors may consider listing their results as well for a comparison. Otherwise, it is hard to tell whether this paper has followed the standard way for evaluation on QM9 and whether the evaluation is meaningful and fair.

2. Some evaluation results are missing. For example, in Table 2, the results for Rand Int and GraphCL are shown, yet PropPred is not shown. Is there a specific reason for this? Similarly, in Table 4, the results of Dist-pred and Conf-gen are missing.

3. What does PropPred mean in the paper? The paper states that "we pre-train baselines by either predicting the properties of GEOM-Drugs’ pre-training subset (labeled Prop Pred)." Which baselines are referring to? What do the properties of GEOM-Drugs correspond to here?

4. Standard deviation is missing in Tables 1, 2, and 3.

5. The settings for baselines are unclear.  Which GNN is used for the baselines like GraphCL? How many layers are used?

6. The paper claims that conformation methods are slow and shows a previous method that takes 6 hours/molecule for conformation. I am not sure whether this claim is correct. Are neural conformers (e.g. https://arxiv.org/abs/1904.00314) as slow as these methods? In my opinion, the inference of neural models shall be relatively faster.

Minor point:

I think the method has some similarities with knowledge distillation. The model takes a 3D GNN as 'teacher', while the 2D GNN learns from the 3D GNN to obtain latent 3D information. Clearly, the methods proposed in this paper have differences from knowledge-distillation-based methods. The paper might discuss these differences and compare with a knowledge distillation baseline if possible.

Typo:

Incomplete sentence: Including 3D molecular structure as input to learned models their performance for many molecular tasks.

**Summary Of The Paper:**

This paper proposes a 3D pretraining method for molecular property prediction. As 3D information is infeasible to compute at the scale required by real-world applications, this paper reasons about the geometry of molecules given only their 2D molecular graphs.

During pretraining with molecules whose 3D information is known beforehand, this paper uses a 2D GNN to encode these molecules. Then, it maximizes the mutual information between 3D summary vectors and the encoded representations for injecting 3D information into the representations. During fine-tuning, the model can take 2D molecules as input. The pretraining phase ensures that the representations during fine-tuning contain latent 3D information.

The paper pre-trains on three datasets, QM9, GEOM-Drugs, and QMugs, and tests on both quantum mechanical properties (QM9 and GEOM-Drugs) and non-quantum properties (10 datasets, e.g. HIV and BACE). The paper claims that significant improvements are obtained for quantum mechanical properties. Also, the method does not suffer from the negative transfer.

**Summary Of The Review:**

The method proposed in this paper is sound. However, the evaluation has some flaws as discussed in the main review section.

I will consider raising my scores if these questions are properly solved during the discussion period.

---

> ### Author Response · Authors · 2021-11-13
> **We included the requested experiments and all other points of the reviewer**
>
> We thank the reviewer for the positive feedback. We are glad that the reviewer found our method to be “novel and sound”, and thinks that our “experiments are well designed”, and that the “paper is clearly written and easy to follow”. The reviewer also suggests three additional experiments which we now included, finds some unclarities in our experiment descriptions, and is concerned about the time it takes to generate conformers for use as explicit 3D information. We address all these topics below and make the changes to the paper together with the requested experiments.
>
> **Our changed paper is available in the Rebuttal Revision that is uploaded to Open Review**
>
> ---
> **The reviewer determines that in Table 4, the results of Dist-pred and Conf-gen would be interesting**
>
> We thank the reviewer for this suggestion and included the results which show that Conf-gen pre-training sometimes beats 3D Infomax, but our method is still the only one which never suffers from negative transfer.
>
> |          | Dist-pred | Conf-gen |
> |----------|-----------|----------|
> | esol     | .947      | .867     |
> | lipo     | .718      | .757     |
> | freesolv | 2.48      | 2.42     |
> | bace     | 76.5      | 80.0     |
> | bbbp     | 66.0      | 66.1     |
> | tox21    | 73.8      | 75.2     |
> | toxcast  | 61.5      | 64.7     |
> | clintox  | 55.7      | 64.2     |
> | sider    | 57.1      | 56.3     |
> | hiv      | 75.6      | 76.5     |
> `Please find the complete Table 4. in the revised paper`
>
>
> **The reviewer points out that there are faster methods for generating conformers than the one we cite for generating high accuracy ones**
>
> There are indeed neural (and classical) models that are much faster. The conformers generated by them are less accurate. However, using them might still yield an improvement and we did not show how this compares against our method. We now include this important comparison in the paper and thank the reviewer for the suggestion.
>
> `Please find these results in column “RDKit SMP” in Table 1 of the revised paper or in our main review response`
>
> We find that while using the fast but less accurate conformers of RDKit increases performance for some properties, it also decreases the performance of others. Our 3D Infomax approach is the only method that consistently provides large improvements for every property.
>
>
> **The reviewer points out a lacking description of the PropPred baseline**
>
> We now elaborate that PropPred refers to pre-training by predicting the Gibbs free energy of the same 120k molecules of GEOM-Drugs that our method uses for pre-training. We see that the quoted sentence was unclear and rephrased it.
>
> **Q: Why is PropPred not shown in Table 2?**
>
> This is the case since PropPred is pre-trained by predicting the Gibbs free energy of the same dataset whose Gibbs free energy is predicted in Table 2. This means we would otherwise pre-train with the same task that is used for finetuning.
>
> **The reviewer points out that we missed to state the featurization used for QM9 and GEOM-Drugs and is concerned about the evaluation settings for QM9**
>
> We added the missing description to the paper. Our evaluation mainly differs from the standard in that we only use 50k (of 134k) molecules for training the models. This is done since the other half of QM9 is sometimes used for pre-training. We would like to argue that the comparison in Table 1 is fair since all methods use the same featurization, GNN, and data. The only exception is the “True 3D SMP” column which applies to the different use case where explicit 3D information is available. Our description of this matter was lacking and we improved it in section 5.1.
>
>
> **Q: Which GNN is used for the baselines like GraphCL? How many layers are used?**
>
> All baselines use the **exact same GNN** architecture as our method which we now explicitly state in the main text of the paper. All hyperparameter settings and how they were selected is detailed in Appendix B.
>
> **The reviewer is interested in the standard deviations for Tables 1, 2, and 3**
>
> For many experiments, the reported values are already averaged over multiple runs and we have the standard deviations but did not put them into the tables since Table 1 and 3 become too wide for the page and to save space. We have now included additional standard deviations in Appendix B.4 and hope that showing the standard deviations for the RandInit column in the main text is helpful to have one deviation for each property directly visible. We are also computing further standard deviations and will include them for e.g. Table 2.
>
> **The reviewer points out a similarity to knowledge distillation**
>
> We think this is a very interesting and intuitive perspective to explain our method and include it in our paper.
>
> ---
> We appreciate the important experiment suggestions and points that improved the paper. We hope that the new results and explanations added to the paper can increase the reviewer's confidence in the significance of the paper.

---

> > ### Comment · Reviewer_7nK8 · 2021-11-16
> > **Thank the authors for the responses.**
> >
> > Some additional comments:
> >
> > (1) Could the authors also provide some results using the full QM9 dataset for training? The authors can exclude the models that need training on QM9. Some comparisons with previous methods on OGB will be also helpful. This shall be a good sanity check.
> >
> > (2) How accurate is RDKIT SMP? Is this the state-of-the-art method for molecule conformation? Does the choice of neural/non-neural conformers greatly affect the final performance?

---

> > > ### Author Response · Authors · 2021-11-20
> > > **Thank you. Our updates and results:**
> > >
> > > We thank the reviewer for the additional experiment suggestions. We provide the results below, and answer the reviewer's questions. We hope these can increase the reviewer's confidence in our work.
> > >
> > > ---
> > > **Q: How accurate is RDKit SMP? Is this the state-of-the-art method for molecule conformation?**
> > >
> > > In a benchmark of conformer generators [1] RDKit shows competitive performance with the other methods of similar speed. RDKit’s distance geometry algorithm is a widely used standard for conformation generation and, e.g., GeoMol [2] compare against it. GeoMol (from June 2021) is the current state-of-the-art neural method for molecular conformation generation. In the next point, we evaluate GeoMol’s conformers as input for a 3D GNN.
> > >
> > > ---
> > > **The reviewer suggests also trying a 3D GNN using lower-cost neural conformers and finding out if “the choice of neural/non-neural conformers greatly affects the final performance”**
> > >
> > > We use the current best neural method for conformation generation GeoMol [2] to generate the conformers of QM9 and use them as 3D information for the 3D GNN “Spherical Message Passing” (SMP) [3]. Since GeoMol does not work for molecules without a dihedral pattern we omit those. Upon visual inspection of the generated conformers, we find that they often exhibit very poor quality if there is a ring in the molecule as can be seen here: https://anonymous.4open.science/r/geomol-conformers. Such outliers are not present in RDKit’s conformers. For the effect on the final performance, we find that these low-quality conformers greatly diminish the performance of SMP. We discuss these results in more depth in the added Appendix C.6.
> > >
> > > |       | RDKit + SMP | GeoMol + SMP |
> > > |-------|-------------|--------------|
> > > | mu    | .4344       | .6046       |
> > > | alpha | .3020       | .8435       |
> > > | homo  | 82.51       | 195.0        |
> > > | lumo  | 80.36       |   201.4   |
> > > | gap   | 114.24      | 284.1        |
> > > | r2    | 22.63       | 65.84        |
> > > | ZPVE  | 5.18        | 17.40        |
> > > | cv    | .1419      | .5467       |
> > > `Please find the complete Table 17. in the paper`
> > >
> > > ---
> > > **The reviewer determines that it would be interesting to compare previous standard architectures used on OGB with the MPNN we employ**
> > >
> > > We provide the comparison of the MPNN architecture that we employ (PNA [4]) with the GIN and GCN results for the OGB datasets that we evaluate on. As can be seen from the table, with the exception of the clintox dataset, PNA performs best or close to best in all benchmarks.
> > >
> > > |           | PNA   | GCN   | GIN   |
> > > |-----------|-------|-------|-------|
> > > | esol↓     | 0.947 | 1.114 | 1.173 |
> > > | lipo↓     | 0.739 | 0.797 | 0.757 |
> > > | freesolv↓ | 2.233 | 2.640 | 2.755 |
> > > | bace↑     | 78.13 | 79.15 | 72.97 |
> > > | bbbp↑     | 68.27 | 68.87 | 68.17 |
> > > | tox21↑    | 73.88 | 75.29 | 74.91 |
> > > | toxcast↑  | 63.62 | 63.54 | 63.41 |
> > > | clintox↑  | 58.98 | 91.30 | 88.14 |
> > > | sider↑    | 55.95 | 59.60 | 57.60 |
> > > | hiv↑      | 77.06 | 76.06 | 77.07 |
> > > Arrows indicate whether higher or lower quantities are better.
> > >
> > > ---
> > > **The reviewer points out that it would be interesting to know how the tested architectures will perform when trained on all of QM9**
> > >
> > > We trained the same Principal Neighborhood Aggregation (PNA) [4] architecture that is used in all QM9 experiments on the full QM9 dataset (100k) instead of on only half of QM9. Similarly, we also trained SMP [3] on all of QM9 using the exact ground truth conformers that are highly computationally expensive to obtain. Below, we report the mean errors of the different methods. As expected, the performance increases with more data but the qualitative outcomes remain the same. The reason why we did not include the results on the full QM9 dataset is that we are simulating a real-world scenario where the dataset from which 3D structures are obtained is mostly disjoint from the dataset with the property of interest.
> > >
> > > |       | PNA all | PNA 50 | SMP all | SMP 50 |
> > > |-------|---------|--------|---------|--------|
> > > | mu    | .3600   | .4133  | .0439   | .0726  |
> > > | alpha | .3955   | .3972  | .0957   | .1542  |
> > > | homo  | 70.67   | 82.10  | 40.11   | 56.19  |
> > > | lumo  | 71.58   | 85.72  | 28.75   | 43.58  |
> > > | gap   | 101.    | 123.08 | 60.02   | 85.10  |
> > > | r2    |         | 22.14  | 0.8531  | 1.51   |
> > > | ZPVE  | 13.76   | 15.08  | 1.9635  | 2.69   |
> > > | cv    | .1680   | .1670  | .0359   | .0498  |
> > >
> > >
> > >
> > > ---
> > >
> > > [1] Friedrich et al. Benchmarking Commercial Conformer Ensemble Generators.
> > >
> > > [2] Ganea et al. GeoMol: Torsional Geometric Generation of Molecular 3D Conformer Ensembles.
> > >
> > > [3] Spherical Message Passing for 3D Molecular Graphs https://openreview.net/forum?id=givsRXsOt9r
> > >
> > > [4] Corso et al. Principal Neighbourhood Aggregation for Graph Nets

---

> > > > ### Comment · Reviewer_7nK8 · 2021-11-21
> > > > **Thanks for the additional experiments.**
> > > >
> > > > First, thanks for the additional experiments.
> > > >
> > > > I will raise my score to 5. There are still some concerns about non-quantum properties since the model seems to underperform Conf-gen on most properties.

---

> > > > > ### Author Response · Authors · 2021-11-22
> > > > > **Thank you for the constructive collaboration to improve the paper.**
> > > > >
> > > > > We would like to note that "Conf-gen" is another 3D pre-training approach proposed in our paper. We see its good performance on some of the OGB datasets as further evidence that pre-training with 3D information provides significant improvements and is a valuable approach.
> > > > >
> > > > > When comparing 3D Infomax to Conf-gen, one can note that the former always performs better on quantum properties. For the OGB datasets, they are complementary with one method often performing well when the other does not provide improvements. Both of our 3D pre-training methods have their merits.
> > > > >
> > > > > We agree that this matter was not discussed in the paper and resolve this in Section 5.3.

---

### Official Review · Reviewer_Awh3 · 2021-11-02

**Correctness:** 3
**Technical Novelty And Significance:** 3
**Empirical Novelty And Significance:** 2
**Recommendation:** 3
**Confidence:** 5

**Main Review:**

Strengths:
This paper starts with an interesting direction, to use 3D geometry to augment the 2D representation. This is an interesting and well-motivated problem.

-----

Weaknesses:

(1) The empirical result is the most important point, but it has the following problems.
1. Motivation contradiction. The empirical result does not match the motivation. In the abstract, it says that `fine-tuning on molecules with unknown geometry, …`; in the intro, it says `After pre-training, … fine-tuned on molecular datasets where no 3D information is available.` However, in Sec 5.1, it is tested on `50k molecules from QM9 or 140k from GEOM-Drugs`, where both include 3D information. Besides, as we can tell from the last column of Table 1, 3D geometric GNN is overwhelmingly better, so it is not quite clear why authors want to test on this dataset. Similar concerns on Table 2-3.
2. The empirical results are confusing. If we ignore the motivation contradiction issue, then according to Method section, this work pre-trains one 2D GNN and one 3D GNN, and downstream tasks include both 2D and 3D information. However, in Table1-3, the authors only check the 2D fine-tuning. If the main focus is on performance improvement, why not focus on 3D downstream tasks? If the performance is about 2D downstream tasks, then why consider these tasks with 3D geometry available? Same concerns on Table 2-3.
3. Performance not good enough and not enough SSL baselines. As pointed out by authors in Intro and Related Work, there are many SSL works along this research line, like AttrMasking, ContextPred, JOAO. GraphCL is more recent than the previous two, but JOAO is more recent. Besides, GraphCL does not show an overwhelming best performance on these tasks, and instead, it shows better performance on different types of graph data. So I’m just wondering why authors didn’t try this? And Table 4 result shows that 3D InfoMax has very limited improvement, or sometimes has even worse performance. This fails to support the usefulness of 3D Infomax.
4. Confusing results. Table 1 result is confusing. It doesn’t specify which datasets are used for the pre-training baselines.

(2) Some descriptions are not clear and terminologies should be specified more clearly.
1. GNN can be adopted on both 2D and 3D graphs, which is not clearly specified in the paper. For example: the abstract says `GNN still generates implicit 3D information …`, and the GNN here refers to the 2D GNN.
2. In Intro, `We pre-train a GNN to encode implicit 3D information …`. This expression is not clear. A better way would be: This work is training 2 GNN (one 2D GNN and one 3D GNN) at the pre-training stage; with MI (SSL), the 2D representation can encode implicit 3D information. Then only 2D GNN is adopted for fine-tuning because downstream tasks have no geometry info.
3. In Intro, `This way, the GNN learns to generate latent 3D information using only the information given by the 2D molecular graphs.` The GNN here also refers to the 2D GNN.
4. The authors mention `2D GNN generate 3D information` at multiple places, like in Intro and Method. But the main solution is to adopt InfoMax, which is a contrastive SSL method; the word `generate` typically implies generative SSL [1]. Authors might better tune these words carefully or explicitly specify them in the paper.

Without specifying these terminologies explicitly and clearly (e.g., at the beginning of the paper), it is hard for readers to follow the logic of this work.

(3) Some claims about the related work in this paper are too bold.
In Intro, `It is not clear why these learned representations should be informative and generalize well.` But all the papers, AttrMasking, ContextPred, GraphLoG, GraphCL, JOAO have claimed the intuitions of the proposed methods, some are supported with ablation studies.
I acknowledge that 3D info is helpful for 2D, but the above claims on existing works are not correct.

(4) Some text words in Fig 1, Fig 2, Fig 3 seem to be hand-written? If so, authors may as well update them in a more formal way.

(5) The 3D Net / 3D GNN representation is wrong.
In Sec 4.1, the equation on 3D Net / 3D GNN is wrong. The representation should be f^b(V, R), i.e., the input to 3D GNN includes both the atom types and atom positions. This holds for SchNet, DimeNet, EGNN, SphereNet.

(6) The objective function is wrong.
The NTXent loss doesn't seem correct to me.
The authors can verify this from the definition of Cross-Entropy, and here I put a more straightforward explanation.
In eq(1) in [2], the numerator is the positive pair; the denominator includes 1 positive pair and 2(N-1) negative paris, i.e., 2N-1 pairs in total.
In eq(1) of this paper, however, the numerator is the positive pair, and the denominator only includes N-1 negative pairs.
Thus, this is definitely not NTXent (similar for the following objective functions).

I haven’t found the codes attached in SI, so I just played around with it on my own implementation, and sometimes (on some datasets) it can lead to negative loss. I’m not sure which one is used in this work, and maybe it can still have some positive effects on downstream tasks, but definitely not NTXent loss.

-----
Minor points:

(1) Sec 4.2 can be moved to Sec 5.

-----
[1]  Liu, Xiao, et al. "Self-supervised learning: Generative or contrastive." IEEE Transactions on Knowledge and Data Engineering (2021).

[2] Chen, Ting, et al. "A simple framework for contrastive learning of visual representations." International conference on machine learning. PMLR, 2020.


**Summary Of The Paper:**

This paper starts with an interesting direction, to use 3D geometry to augment the 2D representation. From the technical point of view, it adopts SSL to maximize the representations of 2 views, so as to enable the 2D GNN to encode 3D geometry information, which can be beneficial for downstream tasks.


**Summary Of The Review:**

This paper is an empirical work, so the empirical results are the most important. However, as listed above, there are some key issues with the empirical performance (point 1). In addition, there are also some other concerns: the writing issues (point 2&3), visualization issues (point 4), and equation issues (point 5-6).

---

> ### Author Response · Authors · 2021-11-13
> **We clarify the use case of our method and include the reviewers suggestions in the paper**
>
> We thank the reviewer for the thorough review. The reviewer seemed to appreciate the paper addressing "an interesting and well-motivated problem", but raised some concerns. Especially with respect to the fact that a 3D GNN using highly accurate 3D structures outperforms 3D Infomax and that these are available for two datasets we evaluate on. In our response, we highlight how we do not use the dataset's 3D structures to simulate the use case of our method is when this 3D information is not available since it is computationally infeasible to obtain. Below, we respond in more detail to all points and incorporate them in our paper which we hope can increase the reviewer's confidence in the significance of our work.
>
> **Our revised paper is uploaded to Open Review**
>
> ---
> **The reviewer points out that we discuss “fine-tuning on molecules with unknown geometry” and then fine-tune on GEOM-Drugs and QM9 which both include 3D information**
>
> When we fine-tune on subsets of the datasets, we do not use their 3D information and act as if it were not available for the purpose of evaluating our method. There are many applications (large-scale screening) where obtaining the structures is computationally infeasible and we simulate this scenario. We now make this more explicit in the paper.
>
> **The reviewer is also concerned about our SSL baseline “GraphCL” in context with “JOAO”**
>
> We see that we described this baseline too imprecisely. JOAO [1] is a framework for automatically selecting the best augmentations for GraphCL. We use the augmentations that the JOAO authors found to work particularly well for OGB's molecules. We now made this more explicit in our paper.
>
> **Another point raised by the reviewer is that we only fine-tune the 2D GNN even though we also employ a 3D network during pre-training**
>
> We think our description of the following was confusing and improved it. While a 3D network is used during pre-training, the aim is not to also fine-tune the 3D network. It only exists for the purpose of pre-training the 2D GNN. The reason for this is explained in the next paragraph.
>
> **The reviewer sees our description of our 3D network as incorrect and that “the input to 3D GNN includes both the atom types and atom positions”**
>
> We agree that this is correct for a standard 3D GNN. Our phrasing caused the confusion that we would use such a 3D GNN and also pre-train it. Unlike SchNet, DimeNet, EGNN, or SphereNet, our 3D network does not take atom features as input. The reason is that we want the 3D network to only extract information about the 3D structure so that the 2D network is forced to do the same to maximize the mutual information. This reason was missing in our paper and we now include it.
>
> **The reviewer determines that some “descriptions are not clear and terminologies should be specified more clearly”**
>
> We see that many statements were unclear with the confusion that a 3D GNN was also being pre-trained in our method. We hope that our improved explanation is now clearer and we no longer use confounding terminology like “generate”.
>
>
> **The reviewer points out missing details in the description of the pre-training methods in Table 1**
>
> We appreciate the reviewer for making us aware of this and we add the details to our description in Section 5.1.
>
> **The reviewer points out that our claim that “It is not clear why these learned representations should be informative and generalize well” is too bold**
>
> We agree with this assessment and think our statement was imprecise and removed it.
>
>
>
> **The reviewer has concerns regarding the NTXent loss**
>
> We agree and make the necessary changes. Our loss is different from NTXent as introduced by SimCLR [2] (eq. 1) in the number of negative samples used. The original loss:
>
> $-log\frac{exp(sim(z_i,z_j)/\tau)}{\sum_{k=1}^{2N} \mathbb{1}_{[k \neq i]} exp(sim(z_i,z_k)/\tau)}$
>
> uses 2(N-1) negative pairs in the denominator with batchsize N.
> Meanwhile we only have N-1 negative samples:
>
> $-log \frac{exp(sim(z_i,z_i)/\tau)}{\sum_{\substack{k=1 \\\\ k\neq i}}^{N}exp(sim(z_i,z_k)/\tau)}$
>
> The denominator does not include the positive sample, which is the reason why negative values are possible. Comparisons with different losses are in Appendix C3. Since our loss is different, we now write that our loss is only "a similar loss" to NTXent and no longer use that term.
>
> **Q: Why is some text in the figures handwritten?**
>
> The figures have been enthusiastically approved by an ICLR general chair (Prof. Deisenroth) prior to submission. If the reviewer thinks the style is not acceptable, we will redo the figures.
>
> **The reviewer could not find our implementation**
>
> The original submission includes an anonymized GitHub repository in the Reproducibility Statement and Experiments section. We hope this can be helpful to the reviewer.
>
> ---
> [1] You et al. Graph contrastive learning automated.
> [2] Chen et al. A simple framework for contrastive learning of visual representations.

---

> > ### Comment · Reviewer_Awh3 · 2021-11-17
> > **Reply to authors**
> >
> > First I appreciate the authors' reply. However, two of my biggest concerns remain unsolved.
> >
> > 1. For downstream with 2D info. I have following concerns.
> >     - The results in Table 1 and 3 are still not quite convincing to me, especially considering that 3D info is available (which contradicts the assumption in the paper). I do get that the authors only use the 2D part for fine-tuning, but since now we know where the optimal performance can be (using SphereNet) and SSL with 2D are orders of magnitudes worse, from the empirical point of view, I don't think this is promising. More related discussions are attached in the second point.
> >     - Table 4 is more important, while the improvements are incremental: the 3D InfoMax ranked 1st in only 1 out of 10 tasks. QM9 is a `quantum mechanics` dataset, and the 10 tasks from Table 4 cover more broader range: `physiology`, `physical chemistry`, and `biophysics`. (Categories come from MoleculeNet [1]) Thus, this somehow shows that 3D InfoMax can only work on one out of four types of tasks, which is not a promising result either.
> >     - This is a minor point. The Table 1 can be organized better. For the current version, the SSL methods and datasets are mixed-up in the head.
> >
> > 2. For downstream with both 2D and 3D. There remain several concerns.
> >     - During pre-training, authors have both 2D and 3D GNN pre-trained, so why don't the authors do fine-tuning with the 3D GNN?
> >     - The current description of 3D GNN has fixed misunderstandings. And I also tried several 3D GNN models (including SphereNet) without any atom features, the performance stay almost the same on 12 QM9 tasks. In other words, the current implementation of 3D GNN can be fine-tuned on QM9 downstream tasks.
> >     - BTW. There are 12 tasks in QM9, can the authors help explain where are the missing four?
> >
> > 3. Plus there are some key issues in the original submission, (though have been solved by authors) I would like to keep my score the same.
> >
> > -------
> > [1] Wu, Zhenqin, et al. "MoleculeNet: a benchmark for molecular machine learning." Chemical science 9.2 (2018): 513-530.

---

> > > ### Author Response · Authors · 2021-11-20
> > > **Thank you. Our further updates and comments:**
> > >
> > > We are glad that many of the reviewer’s concerns have been resolved and address the two remaining ones and their aspects below. We hope these can increase the reviewer's confidence in the impact and relevance of our work.
> > >
> > >
> > > ---
> > > **The reviewer argues that our results are not significant since the 3D method SphereNet (SMP) [1] outperforms our 2D method on QM9**
> > >
> > > We do not aim to establish a new state-of-the-art for QM9. Our method considers the scenario when no 3D structures are available. It is expected that a method such as SphereNet with access to accurate conformers outperforms an approach that does not use 3D information. SphereNet can be considered as an upper bound for the performance that can be reached with 3D Infomax since the implicit 3D information that our method produces from 2D inputs cannot be more informative than the ground truth conformers. \textbf{We argue that our method is important since, on average, it improves the performance of a 2D method by 22% for quantum mechanical properties} by pre-training it. Moreover, as we mentioned in the overall response, we have now added experiments where a SOTA 3D networks like SMP is run on the 3D structures generated with fast methods such as the popular RDKit and the SOTA neural conformer generation method GeoMol. In both cases, using 3D InfoMax provides significantly better results. While this is not important for reaching a new SOTA on QM9, it is relevant for many real world applications where only 2D information is available since it is not feasible to obtain the 3D structures.
> > >
> > > ---
> > > **Q: During pre-training, the authors have both 2D and 3D GNN pre-trained, so why don't the authors do fine-tuning with the 3D GNN?**
> > >
> > > We now performed this test as described by the reviewer. We use a SphereNet (SMP) as 3D network during pre-training and then fine-tune it to predict QM9’s properties with QM9’s accurate conformers as input. We note that this does not address the problem that we are trying to solve in this paper, where no 3D information is available. However, we agree that the results can be interesting and provide them below and in the paper in Table 16. We find that the reviewer's suggested method indeed improves the 3D GNN's performance even though it does not have access to the atomic features. This may be due to the covalent bonding structure and other 2D information edge information that is available during pre-training and which SMP usually cannot use since it employs a radius graph. This could be an interesting future direction to attempt beating the state-of-the-art for quantum mechanical property prediction when accurate 3D conformers are available, which we mention in Appendix C.5.
> > >
> > > |       | SMP    | SMP pre-trained |
> > > |-------|--------|-----------------|
> > > | mu    | 0.0726 | 0.0801          |
> > > | alpha | 0.1542 | 0.1276          |
> > > | homo  | 56.19  | 44.50           |
> > > | lumo  | 43.58  | 37.48           |
> > > | gap   | 85.10  | 70.45           |
> > > | r2    | 1.51   | 1.12            |
> > > | zpve  | 2.69   | 2.43            |
> > > | cv    | 0.0498 | 0.0421          |
> > >
> > >
> > > ---
> > > **Q: Why did we only evaluate on eight of QM9’s properties?**
> > >
> > > We were initially following the evaluation of the SE(3)-Transformer [2], which omitted the other quantities. We are currently computing the additional results and will include them in the paper.
> > >
> > > ---
> > > **The reviewer sees the results for OGB as more relevant than the results for QM9**
> > >
> > > We agree that our method does not provide outstanding improvements for the OGB datasets. We think it is still an interesting approach and worth trying since it sometimes yields improvements but never experiences negative transfer where the downstream performance is decreased by pre-training as it sometimes happens with other SSL methods. However, for the quantum mechanical properties of QM9, we always observe a large improvement over the 2D method that was not pre-trained, and we think that the method is, therefore, useful for this domain.
> > >
> > > ---
> > > [1] Spherical Message Passing for 3D Molecular Graphs https://openreview.net/forum?id=givsRXsOt9r
> > >
> > > [2] Fuchs et al. SE(3)-Transformers: 3D Roto-Translation Equivariant Attention Networks

---

### Official Review · Reviewer_SAmw · 2021-11-03

**Correctness:** 3
**Technical Novelty And Significance:** 3
**Empirical Novelty And Significance:** 2
**Recommendation:** 8
**Confidence:** 2

**Details Of Ethics Concerns:**

Non, ethical concerns are given in the paper.

**Main Review:**

I found the introduction and background good to read and easy to understand. The paper is supported by great images to explain what was done. It also admits the weakness of the method with biological systems. The non-QM property part is not as clearly written and I found it harder to understand which dataset belongs to which statement.
-	Page 4 line 5: I believe the statement that it learns quantum mechanical (QM) interactions is not backed up. The 3D structures of molecules are partially determined by steric hindrance and not purely by QM interactions. E.g. when the model is pretrained on a smaller set of elements and used for elements it has not seen this shows it learned the structure and not QM.
-	Page 3 line 3: What is SE(3) symmetry?
-	Table 1: Is there a reason why two properties (cv and alpha) are better predicated with the pretraining of QMugs than with the same dataset (QM9)?

Some very minor points:
-	For the review is would be with advantage to add a column with line numbers. Further, exchanging numbers for the citations instead of name and year, this improved the fluent reading experience.
-	Page 1 figure 1: MI could be mentioned in the Figure comment or before on page 1 paragraph 4 (our solution) as this abbreviation was not used yet.
-	Abstract, Page 2 line 4, page 6: Inconsistency with the number of quantum mechanical properties. 8 are given in the tables but often 10 are mentioned.
-	Page 2 last line: Year is missing in the citation.
-	Page 4 last line: The word hardest is confusing here.
-	Page 7 Table 1: The targets are never mentioned or explained. Only the units are given in the SI. Table 6, here also the names of the variables would be great.
- Page 8 after table 4, last sentence of the paragraph: Incomplete and I do not understand it.
-	Page 10ff: The citations are not normed, e.g. PMID


**Summary Of The Paper:**

The paper uses the 3D structures of molecules to pre-train graph neural networks (GNN) representation of 2D molecules. This improves the molecular property prediction of some quantum mechanical (from 8 tested 8 were improved) and non-quantum mechanical properties (from 10 tested 4 were improved).

**Summary Of The Review:**

For molecules the 3D information is essential and this paper shows an affordable method to include this for improved property prediction. There are two main advantages, firstly it can be used for a dataset where 3D information is not available and secondly it is cheaper than implementing 3D information in the representation.

---

> ### Author Response · Authors · 2021-11-13
> **We make all changes mentioned in the detailed review**
>
> We thank the reviewer for the time taken to thoroughly review our work. We are glad that the reviewer found the paper “good to read and easy to understand" and was persuaded by the ideas and the results presented. Below we answer the questions that were raised and we make all the desired changes in our paper.
>
> **Our changed paper is available in the Rebuttal Revision that is uploaded to Open Review**
>
> ---
> **Major Bullet Points**
> 1. **The reviewer raises concerns about our statement “Intuitively, it learns to reason about quantum mechanical interactions”**
>
> We agree with the reviewer's assessment and it is a vague intuition which we therefore remove.
>
> 2. **Q: What is SE(3) symmetry?**
>
> This means being invariant to rotations and translations. We have clarified this point in the paper. SE(3) stands for the Special Euclidean group in 3D.
>
> 3. **Q: Why are cv and alpha better predicted when pre-training with QMugs than when pre-training with QM9?**
>
> In general, we observe that the models generalize well when training on either Drugs or QMugs and transferring to QM9. For the alpha and cv properties, the bigger size of the QMugs pre-training dataset with 620k molecules potentially causes the performance to improve more than the benefit that is obtained from transferring between the same distributions when pre-training with QM9.
>
> Both the alpha (isotropic polarizability) and cv (heat capacity) are highly dependent on the flexibility of the molecule and its movement when subjected to external energy (electromagnetic or heat). It is possible that training on larger molecules and datasets allows the model to better understand flexibility, rotational and vibrational modes, thus transferring better to smaller molecules. However, without empirically studying this in more depth, our explanation remains a hypothesis.
>
> **The reviewer found it unclearly explained which dataset belongs to which statement in the non-QM property part**
>
> We added which datasets we mean in our statements.
>
> ---
> **Minor Points**
> 1. We agree, but have to adhere to the citation style of the ICLR Latex template.
> 2. We now introduce the MI abbreviation in the text before using it.
> 3. The “10 quantum properties” refer to the 8 properties of QM9 plus the 2 properties of GEOM-Drugs which we evaluate on in Table 2.
> 4. We fixed the citation.
> 5. We now explain it to be the negative pair with the highest similarity.
> 6. We now include a brief description of the quantum mechanical properties in Table 6 instead of only stating the units.
> 7. The sentence was confusing: we split it and rewrote it with more clarity.
> 8. The references are now more uniformly formatted. Please let us know if there are important issues.
>
> ---
> We hope that we sufficiently addressed the reviewer’s points regarding correctness and that all statements are now well-supported. Furthermore, we hope that our clarifications can increase the reviewer's confidence in his assessment.

---

### Official Review · Reviewer_2Vfa · 2021-11-04

**Correctness:** 4
**Technical Novelty And Significance:** 3
**Empirical Novelty And Significance:** 3
**Recommendation:** 6
**Confidence:** 3

**Main Review:**

The major strength of this work is the extensive empirical evidence supporting the claims that the proposed self-supervised learning scheme (mutual information between 2D GNN and 3D GNN learned embeddings) significantly improves performance on downstream tasks as compared to other pre-training strategies.  In addition, the language is simple and clear, and the figures provide easily understandable graphical depictions of the method.

The major weakness, which is addressed by the authors, is that these 2D models, even when pre-trained with 3D information, still significantly underperform methods that can leverage 3D information in the downstream task.  This implies that in settings where accuracy is key, one should still opt for the computational expense of generating the 3D conformers directly.  While the authors argue that the gold standard methods of generating 3D conformers can be prohibitively expensive, I would like to see what the performance of 3D GNN operating on conformers such as those that might be generated by a fast method like GeoMol.  This would allow for better evaluation of the tradeoff between accuracy and speed provided by explicit 3D conformer modeling.

**Summary Of The Paper:**

The authors present 3D Infomax, a graph neural network (GNN) pre-training solution that leverages 3D information to generate better learned embeddings and improve performance on down-stream prediction tasks where 3D information would be useful but not easily obtainable.  The approach is useful for a range of downstream tasks involving molecules, including ones that are quantum mechnical, biological, and pharmacological in nature.  They also demonstrate that the use of multiple 3D conformations (thereby encoding the inherent flexibility of molecules) further improves performance.

**Summary Of The Review:**

Overall, I think this is a strong paper due to the compelling results and clarity of communication.  I currently recommend weak accept as I would like to see the experiment I mentioned above (using lower quality, but quick-to-generate 3D conformers) to better understand the tradeoffs involved.

---

> ### Author Response · Authors · 2021-11-13
> **We included the important requested experiment**
>
> We thank the reviewer for the time taken to review our work and constructive feedback provided. We are glad that the reviewer enjoyed the work, appreciates the "extensive empirical evidence" and believes that the presentation is “simple and clear”. We also welcome the evidently thorough understanding of our method and consideration of its use cases which led to an important experiment suggestion (using lower quality, but quick-to-generate 3D conformers). We included these results in the paper which we briefly discuss below.
>
> **Our changed paper is available in the Rebuttal Revision that is uploaded to Open Review**
>
> ---
> **The reviewer identifies that it is important to know the performance of 3D GNNs operating on conformers generated by a fast method**
>
> We agree. For many real-world applications, generating high-quality conformers such as they come with the GEOM-Drugs and QM9 datasets and are commonly used by 3D GNNs is computationally infeasible. This is the scenario where our method is useful as it does not require 3D inputs during inference but still improves performance with the 3D information it generates itself. However, the reviewer correctly points out that conformers can also be generated in a faster way if they are of lower accuracy. These can then be used in a 3D GNN and it is important to know how this compares with our 3D Infomax.
>
> We use RDKit’s distance geometry algorithm ETKDG [1] to generate conformers since it is an established standard tool and use Spherical Message Passing [3] as 3D GNN. We use RDKit instead of GeoMol [2] since GeoMol does not work for molecules without a dihedral pattern which are 20% of molecules in QM9.
>
> |       | RDKit SMP |
> |-------|-------------|
> | mu    | .4344       |
> | alpha | .3020       |
> | homo  | 82.51       |
> | lumo  | 80.36       |
> | gap   | 114.24      |
> | r2    | 22.63       |
> | ZPVE  | 5.18        |
> | cv    | 0.1419      |
> `Please find the complete Table 1. in the revised paper`
>
> We find that while using the fast but less accurate conformers of RDKit increases performance for some properties, it also decreases the performance of others. Our 3D Infomax approach is the only method that consistently provides large improvements for every property and outperforms the RDKit ETKDG baseline on 7 out of the 8 properties. With this comparison, we feel confident in saying that our approach is relevant for applications that require fast inference since it outperforms using low-cost conformers.
>
> ---
> **The reviewer argues that  “in settings where accuracy is key, one should still opt for the computational expense of generating the 3D conformers directly”**
>
> We agree but want to add that for many important applications where large amounts of molecules need to be processed (millions or billions for some virtual screening cases) this computational expense would be really considerable: using CREST (which is still faster than more accurate methods involving ab initio DFT calculations) requires about 6 hours per drug-like molecule per CPU-core [4]. While there are faster methods, the reviewer’s suggested experiments show that using their less accurate conformers does not match our method. We agree that when computationally feasible, high-quality conformers with a 3D GNN should be preferred. But for many common applications, it is not feasible and our method can considerably improve performance when only 2D inputs are available.
> We have improved the discussion of this trade-off in the paper.
>
> ---
> We appreciate the reviewer's important suggested experiment and hope that the new results and considerations increase the reviewer's confidence in the significance of our work.
>
> [1] Greg Landrum. Rdkit: Open-source cheminformatics software
>
> [2] Ganea et al. GeoMol: Torsional Geometric Generation of Molecular 3D Conformer Ensembles.
>
> [3] Spherical Message Passing for 3D Molecular Graphs https://openreview.net/forum?id=givsRXsOt9r
>
> [4] Axelrod et al. GEOM: Energy-annotated molecular conformations for property prediction and molecular generation

---

> > ### Author Response · Authors · 2021-11-21
> > **Additional results of 3D GNNs operating on conformers generated by a learned method**
> >
> > We thank the reviewer again for the proposed experiment of using a 3D GNN with lower quality, but quick-to-generate 3D conformers and the other constructive feedback.
> >
> > ---
> > In the previous comment, we decided to use RDKit [1] to generate conformers in a fast manner instead of GeoMol [2], since GeoMol does not work for ~20% of QM9's molecules. Now we omit those molecules and additionally evaluate using the conformers generated by GeoMol as input for the 3D GNN SMP [3]. Upon visual inspection of GeoMol's conformers, we find that they often exhibit very poor quality if there is a ring in the molecule as can be seen here: https://anonymous.4open.science/r/geomol-conformers. Such outliers are not present in RDKit’s conformers. For the effect on the final performance, we find that these low-quality conformers greatly diminish the performance of SMP. We discuss these results in more depth in the added Appendix C.6.
> >
> > |       | RDKit + SMP | GeoMol + SMP |
> > |-------|-------------|--------------|
> > | mu    | .4344       | .6046       |
> > | alpha | .3020       | .8435       |
> > | homo  | 82.51       | 195.0        |
> > | lumo  | 80.36       |   201.4   |
> > | gap   | 114.24      | 284.1        |
> > | r2    | 22.63       | 65.84        |
> > | ZPVE  | 5.18        | 17.40        |
> > | cv    | .1419      | .5467       |
> > `Please find the complete Table 17. in the paper`
> >
> > ---
> > [1] Greg Landrum. Rdkit: Open-source cheminformatics software
> >
> > [2] Ganea et al. GeoMol: Torsional Geometric Generation of Molecular 3D Conformer Ensembles.
> >
> > [3] Spherical Message Passing for 3D Molecular Graphs https://openreview.net/forum?id=givsRXsOt9r

---

### Author Response · Authors · 2021-11-13
**We summarize the added experiments and improvements that resulted from the reviewers' constructive feedback**

We thank all the reviewers for the time they spent reviewing our work. We are glad that all the reviewers found our work novel and the paper well written and are very committed to improving the paper from their feedback. We integrated every piece of feedback in the new version of the paper.

In the revised paper, the changes are highlighted in green. Our main paper has significant changes. Here are our main highlights:

# Experimental improvements and updates:


* **Additional baseline using cheap conformers** (Table 1): We added the important suggested experiment to evaluate a 3D GNN when using conformers generated with fast methods. These are less accurate than the high-quality conformers of methods such as CREST [1] but more computationally feasible to obtain in the high throughput scenarios where our method is valuable. We employ RDKit’s ETKDG [2] algorithm to generate the conformers and use Spherical Message Passing (SMP) [3] as 3D GNN. We find that this alternative solution for our considered problem setting does not consistently provide improvements and is significantly outperformed by our 3D Infomax method. The results are also in the table of the next point.


* **Fast neural conformers instead of RDKit ETKDG** (Table 17): We use the official implementation of GeoMol [4] (SOTA neural method for conformation generation) to generate conformers of QM9 and show that outlier conformers with particularly poor quality cause SMP [3] to perform poorly. We discuss this in the added Appendix C.6.
|       | RDKit + SMP | GeoMol + SMP |
|-------|-------------|--------------|
| mu    | .4344       | .6046       |
| alpha | .3020       | .8435       |
| homo  | 82.51       | 195.0        |
| lumo  | 80.36       |  201.4   |
| gap   | 114.24      | 284.1        |
| r2    | 22.63       | 65.84        |
| ZPVE  | 5.18        | 17.40        |
| cv    | .1419      | .5467       |
`Please find the complete Table 17. in the paper`

* **Additional 3D pre-training baselines for OGB datasets** (Table 4): As recommended, we computed the results that two 3D pre-training baselines achieve on the Open Graph Benchmark’s molecule datasets and added them to Table 4.
|          | Dist-pred | Conf-gen |
|----------|-----------|----------|
| esol     | .947      | .867     |
| lipo     | .718      | .757     |
| freesolv | 2.48      | 2.42     |
| bace     | 76.5      | 80.0     |
| bbbp     | 66.0      | 66.1     |
| tox21    | 73.8      | 75.2     |
| toxcast  | 61.5      | 64.7     |
| clintox  | 55.7      | 64.2     |
| sider    | 57.1      | 56.3     |
| hiv      | 75.6      | 76.5     |
`Please find the complete Table 4. in the paper`

* **Use Infomax setup to pre-train a 3D GNN** (Table 16): We pre-train SMP [3] on one half of QM9 and fine-tune it on the other half to find that this scheme indeed slightly improves the 3D GNN's performance. This may be due to the covalent bonding structure and other 2D edge information that is available during pre-training and which SMP usually cannot use since it employs a radius graph.

# 3D Infomax Significance
3D Infomax is impactful for predicting molecular properties when only the 2D molecular graph is available. This scenario, e.g., arises when large amounts of molecules need to be processed, such that computationally inferring their precise 3D structures is infeasible. This includes many important real-world applications, such as large-scale virtual screening for drug discovery purposes. For these applications, 3D Infomax can improve the prediction accuracy of a GNN by pre-training it to produce latent 3D information. Performance increases while inference speed remains fast. We highlight this throughout the paper.

# 3D Infomax wide applicability
Our pre-trained models can be integrated to boost the quality of several other applications. While we only evaluate property predictions, 3D Infomax could also be useful in other domains where the 3D structure is unknown. For example, predicting the function of proteins whose 3D structure is much harder to obtain than it is for small molecules.

# 3D Infomax principled
Conventional pre-training methods often learn invariance to augmentations. For molecules, augmentations like dropping an atom often significantly change its properties, and the models could learn incorrect invariances. Meanwhile, 3D Infomax pre-training teaches the model to reason about how atoms interact in space, which is a principled and generalizable form of information.

---
[1] Stefan Grimme. Exploration of chemical compound, conformer, and reaction space with metadynamics simulations based on tight-binding quantum chemical calculations.

[2] Greg Landrum. Rdkit: Open-source cheminformatics software

[3] Spherical Message Passing for 3D Molecular Graphs https://openreview.net/forum?id=givsRXsOt9r

[4] Ganea et al. GeoMol: Torsional Geometric Generation of Molecular 3D Conformer Ensembles.

---

### Decision · Program_Chairs · 2022-01-20

**Decision:**

Reject

**Comment:**

This paper aims to use pre-training to bridge the gap in performance between 2D GNN and 3D GNN. Specifically, during pretraining, it trains both 2D GNNs and 3D GNNs on data equipped with 3D geometry to maximize the mutual information between the 2D GNN representation with the 3D GNN representation. The proposed approach is interesting and novel and the paper presents some promising results showing that the pre-training does provide some benefits for downstream tasks where 3D geometry information is not available in comparison to several other baseline pretraining methods. While the reviewers agree that property prediction without only 2D graph is a practically important setting for high throughput screening, there are concerns about whether the current set of results paint a clear picture on the benefits and superiority of the proposed methods to alternatives (e.g., vs conf-gen) even after the revision. This is not due to lacking of results, but more of a presentation issue where results are not organized and discussed clearly to provide a coherent story.  We do see clear and strong potential for this paper but it needs a careful rewrite/re-organization to tease out the key messages and how the experiments support them.